# Long noncoding RNA licensing of obesity-linked hepatic lipogenesis and NAFLD pathogenesis

Xu-Yun Zhao[1], Xuelian Xiong[1,2], Tongyu Liu[1], Lin Mi[1], Xiaoling Peng[1], Crystal Rui [1], Liang Guo[1], Siming Li[1], Xiaoying Li[2] & Jiandie D. Lin [1]

Hepatic lipogenesis is aberrantly induced in nonalcoholic fatty liver disease (NAFLD) via activation of the LXR-SREBP1c pathway. To date, a number of protein factors impinging on the transcriptional activity of LXR and SREBP1c have been elucidated. However, whether this regulatory axis interfaces with long noncoding RNAs (lncRNAs) remains largely unexplored. Here we show that hepatic expression of the lncRNA Blnc1 is strongly elevated in obesity and NAFLD in mice. Blnc1 is required for the induction of SREBP1c and hepatic lipogenic genes in response to LXR activation. Liver-specific inactivation of Blnc1 abrogates high-fat diet-induced hepatic steatosis and insulin resistance and protects mice from diet-induced nonalcoholic steatohepatitis. Proteomic analysis of the Blnc1 ribonucleoprotein complex identified EDF1 as a component of the LXR transcriptional complex that acts in concert with Blnc1 to activate the lipogenic gene program. These findings illustrate a lncRNA transcriptional checkpoint that licenses excess hepatic lipogenesis to exacerbate insulin resistance and NAFLD.

[1] Life Sciences Institute and Department of Cell & Developmental Biology, University of Michigan Medical Center, Ann Arbor, MI 48109, USA. [2] Ministry of Education Key Laboratory of Metabolism and Molecular Medicine, Department of Endocrinology and Metabolism, Zhongshan Hospital, Fudan University, Shanghai 200030, China. Correspondence and requests for materials should be addressed to J.D.L. (email: jdlin@umich.edu)

Nonalcoholic fatty liver disease (NAFLD) is a prevalent hepatic manifestation of the metabolic syndrome that ranges from simple steatosis to nonalcoholic steatohepatitis (NASH)[1–3]. The latter is characterized by the presence of persistent liver injury, chronic inflammation, and varying degree of liver fibrosis. It is estimated that over a third of the adult population in the U.S. have fatty liver with approximately 5–10% of these individuals further progressing into NASH. Patients with NASH have increased risk for end-stage liver disease such as cirrhosis and hepatocellular carcinoma. NAFLD is rapidly emerging as a leading indication for adult liver transplantation due to its increasing prevalence worldwide and a lack of effective therapies[4]. Several pathogenic mechanisms have been implicated in NAFLD pathogenesis, including insulin resistance, mitochondrial dysfunction, lipotoxicity, and endotoxin exposure[5–8]. Among these, the pathophysiological factors that drive excess liver fat accumulation are considered to play a central role in initiating and perpetuating the cascade of NAFLD pathologies. Recent work has revealed endocrine signaling by adipocyte-derived secreted factors as an important checkpoint for hepatic lipogenesis and NASH pathogenesis[9–11].

Previous studies have established a close association between obesity and aberrant stimulation of hepatic lipogenesis[12,13]. Together with elevated lipid flux as a result of adipose tissue dysfunction, this increase of de novo lipid synthesis aggravates hepatic steatosis in the insulin resistant state. Liver X receptor (LXR) and sterol regulatory element-binding protein 1c (SREBP1c) are central regulators of the hepatic lipogenic gene program[14,15]. Key protein components of the LXR-SREBP1c pathway have been uncovered that mediate transcriptional activation by LXRs and nutrient and hormonal regulation of SREBP1c. However, whether the LXR-SREBP1c axis interfaces with long noncoding RNAs (lncRNAs), an emerging class of metabolic regulators, remains largely unknown.

Similar to protein-coding genes, many lncRNAs exhibit restricted tissue distribution and are tightly regulated by developmental and physiological signals[16,17]. Recent transcriptomic studies have revealed tissue-specific repertoires and regulation of lncRNA expression in adipose tissue and the liver[18–20]. Several lncRNAs, including brown fat-enriched lncRNA 1 (Blnc1) and lnc-BATE1, have been demonstrated to regulate thermogenic adipocyte differentiation, while liver-specific triglyceride regulator (lncLSTR) and LeXis are lncRNA regulators of bile acid and cholesterol biosynthesis, respectively[21,22]. In this study, we demonstrate that Blnc1 is a component of the LXR transcriptional complex that is required for SREBP1c induction and hepatic lipogenic activation in obesity. CRISPR/Cas9-mediated liver-specific inactivation of Blnc1 abrogates HFD-induced hepatic steatosis and insulin resistance and protects mice from diet-induced NASH.

## Results

### Hepatic Blnc1 is elevated in obesity and promotes de novo lipogenesis.
We previously demonstrated that the conserved lncRNA Blnc1 regulates brown and beige adipocyte differentiation and thermogenesis[20,23]. Interestingly, abundant Blnc1 expression was observed in the liver, estimated to be approximately 120 copies per mouse hepatocyte (Supplementary Fig. 1a, b), raising the possibility that it may orchestrate distinct metabolic responses in a tissue-specific manner. To explore this, we first examined whether hepatic Blnc1 expression is altered in diet-induced and genetic obese mice. Quantitative PCR (qPCR) analysis revealed that Blnc1 expression was significantly elevated in the livers from high-fat diet (HFD) fed mice and leptin-deficient (ob/ob) and leptin receptor-deficient (db/db) obese mice (Fig. 1a).

This increased expression of Blnc1 was associated with induction of Srebp1c, a master regulator of lipogenesis, and elevated expression of lipogenic genes, including fatty acid synthase (Fasn) and stearoyl-CoA desaturase 1 (Scd1) (Supplementary Fig. 1c). In a cohort of HFD-fed wild-type (WT) C57BL/6 mice exhibiting different degree of weight gain, hepatic Blnc1 expression strongly correlated with obesity and liver triglyceride (TAG) content (Fig. 1b). The low weight gainers had lower Blnc1 expression in the liver compared to those exhibiting more pronounced adiposity following HFD feeding. These results demonstrate that liver Blnc1 expression is strongly linked to obesity and hepatic steatosis.

Whether lncRNAs play a role in physiological regulation of hepatic lipogenesis and NAFLD pathogenesis has not been established. We next tested the hypothesis that Blnc1 may serve as a lncRNA regulator of hepatic lipid metabolism. Adenoviral overexpression of Blnc1 in primary hepatocytes significantly augmented the induction of Srebp1c and Fasn by T0901317, a synthetic LXR agonist (Fig. 1c). In accordance, the incorporation of $^{14}C$-labeled acetate into lipids was significantly increased by Blnc1 overexpression under basal condition and following LXR agonist treatments (Fig. 1d). We did not observe an effect of Blnc1 overexpression and shRNA knockdown on mechanistic target of rapamycin (mTOR)-Ser/Thr protein kinase AKT signaling in cultured hepatocytes, which acts upstream to promote de novo lipogenesis[24,25] (Supplementary Fig. 2a). We generated a recombinant adeno-associated virus (AAV) expressing Blnc1 under control of the liver-specific thyroid-binding globulin (TBG) promoter to probe its effects on hepatic lipogenesis in mice. Similar to adipocytes[20], Blnc1 is primarily localized in the nucleus of hepatocytes (Supplementary Fig. 2b). Compared to GFP control, mice transduced with AAV-Blnc1 had significantly elevated levels of TAG, but not cholesterol, in their plasma and livers (Fig. 1e). Immunoblotting and qPCR analyses revealed that Blnc1 robustly induced the precursor and the processed nuclear isoforms of SREBP1 and the expression of genes responsible for de novo lipogenesis and lipid storage, including Srebp1c, Fasn, Scd1, Fsp27, Dgat2, and Fabp4 (Fig. 1f, g). In addition, Blnc1 induced mRNA expression of several other LXR target genes (Abca1, Abcg5, Lpcat3, and Rnf145). These results indicate that Blnc1 is sufficient to stimulate de novo lipogenesis in cultured hepatocytes and in vivo in the liver.

### Blnc1 is required for LXR-mediated activation of hepatic lipogenesis.
Our observations that Blnc1 acts in concert with LXR raise the possibility that it may function as a critical lncRNA component of the transcriptional regulatory network that drives hepatic lipogenesis. To determine whether Blnc1 is required for LXR-mediated lipogenic induction in vivo, we generated whole body Blnc1 knockout (KO) mice using CRISPR/Cas9 (Supplementary Fig. 3a–c), and gavaged WT and Blnc1 KO mice daily with vehicle or T0901317 for a total of 4 days. LXR agonist treatment stimulated lipogenic gene expression in the liver, leading to hepatic steatosis and hypertriglyceridemia in mice (Fig. 2a–c), similar to previous findings[26]. Remarkably, LXR agonist-induced rise in plasma TAG and hepatic steatosis was significantly blunted by Blnc1 deficiency. Plasma and liver cholesterol levels were largely unaffected by Blnc1 inactivation (Supplementary Fig. 4a). Hepatic gene expression analysis indicated that Blnc1 inactivation significantly diminished mRNA induction of Srebp1c, Fasn, Scd1, Fsp27, Abca1, and Abcg5 (Fig. 2c and Supplementary Fig. 4b). The induction of SREBP1 protein expression by LXR agonist was also attenuated in Blnc1 KO mouse livers (Fig. 2d). Hepatic lipogenesis is highly inducible in response to feeding. While liver Blnc1

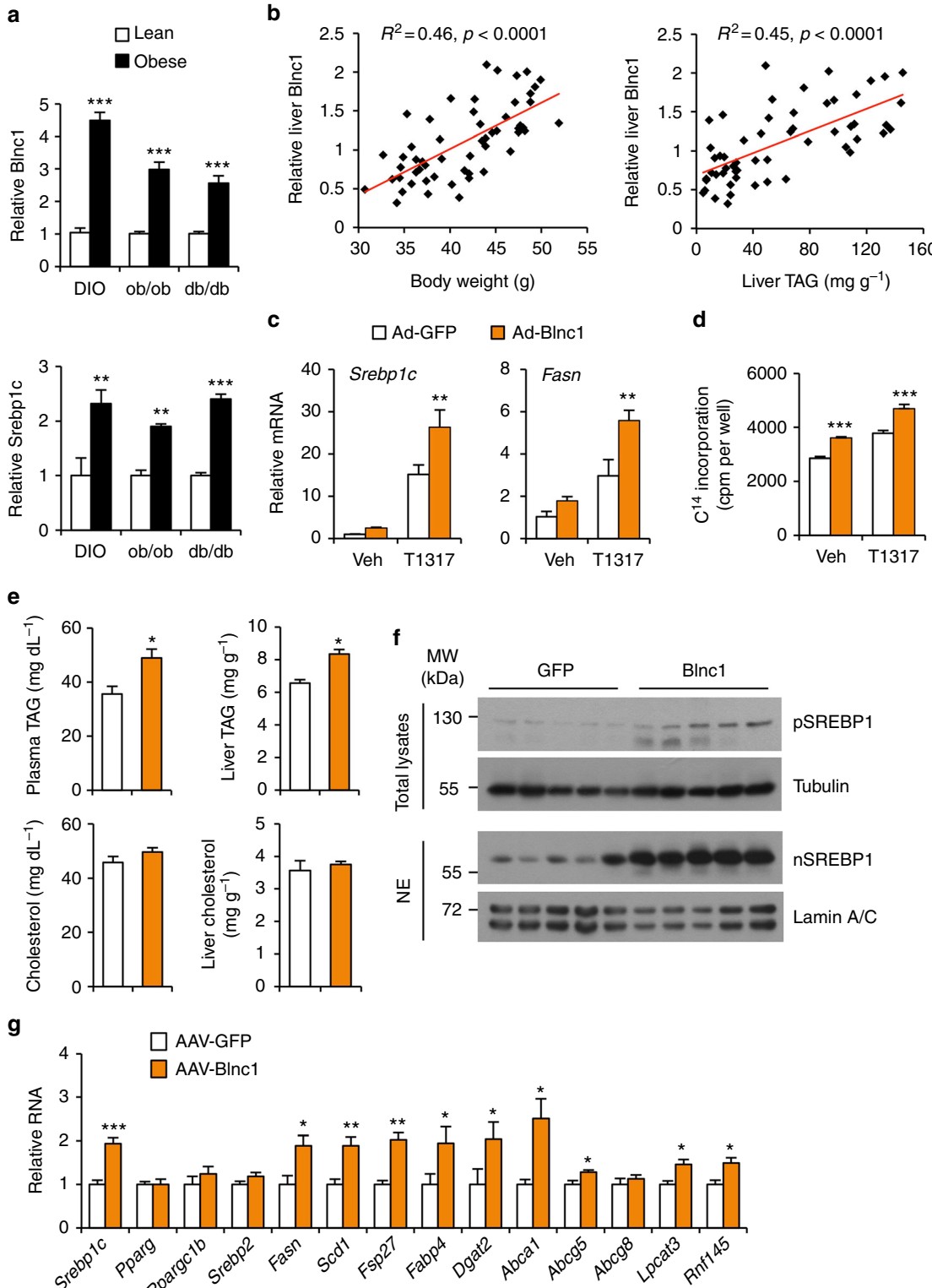

**Fig. 1** Hepatic Blnc1 is linked to obesity and lipogenic activation. **a** qPCR analysis of Blnc1 and *Srebp1c* expression in the livers from lean (open) and obese (filled) mice. For diet-induced obesity (DIO), WT mice were fed chow ($n = 4$) or HFD ($n = 12$) for 10 weeks. For genetic obesity, WT ($n = 5$), ob/ob ($n = 3$), or db/db ($n = 6$) mice were fed standard chow. Data represent mean ± SEM. **$p < 0.01$, ***$p < 0.001$, obese vs. lean, two-tailed unpaired Student's $t$-test. **b** Correlation between liver Blnc1 RNA levels and body weight (left) or liver fat content (right) in HFD-fed mice. **c** qPCR analysis of gene expression in hepatocytes transduced with GFP (open) or Blnc1 (filled) adenovirus followed by treatment with vehicle (Veh) or 5 μM T0901317 (T1317) for 24 h. **d** Incorporation of [14]C-acetate into lipids in transduced hepatocytes treated with Veh or T1317 for 24 h. Data in (**c**) and (**d**) represent mean ± SD ($n = 3$). **$p < 0.01$, ***$p < 0.001$, GFP vs. Blnc1, two-way ANOVA. **e** Plasma and liver TAG and cholesterol in mice transduced with AAV vectors expressing GFP ($n = 5$) or Blnc1 ($n = 7$). **f** Immunoblots of total liver lysates and liver nuclear extracts (NE) from transduced mice. **g** qPCR analysis of hepatic gene expression in transduced mice. Data in (**e**) and (**g**) represent mean ± SEM. *$p < 0.05$, **$p < 0.01$, ***$p < 0.001$, GFP vs. Blnc1, two-tailed unpaired Student's $t$-test

expression remained similar under different feeding conditions, its deficiency impaired refeeding-induced lipogenic gene expression, leading to lower plasma TAG levels (Supplementary Fig. 5).

As Blnc1 is known to regulate brown fat thermogenesis, it is important to establish that it exerts its effects on hepatic lipogenesis through a cell-autonomous mechanism. We

performed LXR agonist treatments in primary hepatocytes isolated from WT and Blnc1 null mice. Similar to in vivo studies, LXR-mediated lipogenic response was significantly impaired in Blnc1-deficient hepatocytes, resulting in lower incorporation of $^{14}$C-labeled acetate into lipids (Fig. 2e, f). These results strongly suggest that Blnc1 and LXR may act in concert to elicit a full

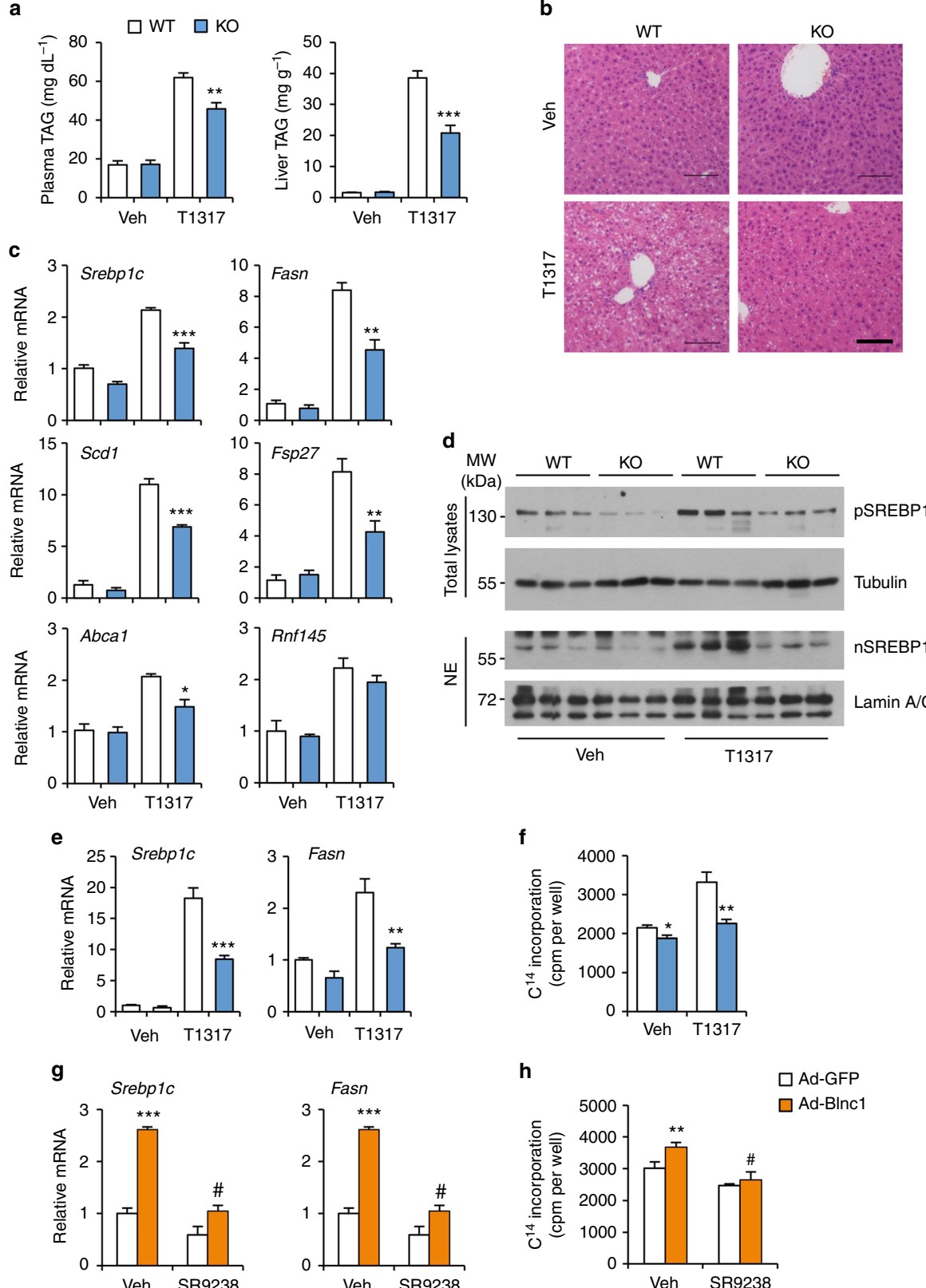

lipogenic response in hepatocytes. In support of this, treatment of hepatocytes with SR9238, a potent and selective LXR inverse agonist[27], significantly diminished the stimulatory effects of Blnc1 on lipogenic gene expression and lipid synthesis (Fig. 2g, h). The reciprocal dependence between Blnc1 and LXR illustrates a critical role of lncRNA-protein crosstalk in governing hepatic lipid metabolism.

**Liver-specific inactivation of Blnc1 protects mice from diet-induced metabolic disorders.** Whole body Blnc1 deficiency likely perturbs energy metabolism in adipose tissue and the liver given its prominent role in the regulation of brown and beige fat thermogenesis. To address this, we developed a new method to generate liver-specific Blnc1 KO (LKO) mice using CRISPR/Cas9. Recently, a transgenic mouse strain constitutively expressing Cas9 was generated for in vivo gene editing in mice[28]. We generated a recombinant adenoviral vector expressing two single-guide RNAs (sgRNAs) designed using a CRISPR design tool (http://crispr.mit.edu/)[29]. We transduced Cas9 transgenic mice with this sgRNA adenoviral vector to specifically ablate Blnc1 in the liver (Fig. 3a). As expected, adenoviral sgRNA expression resulted in liver-specific inactivation of Blnc1 without affecting mRNA expression of the adjacent gene progesterone and adipoQ receptor family member 9 (*Paqr9*) (Supplementary Fig. 6a–c). We observed that Blnc1 expression was reduced by approximately 55% in the liver, but not in other tissues, such as adipose tissue, skeletal muscle, and the brain. We suspected that this moderate reduction of Blnc1 might be due to its expression in non-parenchymal cells (NPC) in the liver, which are poorly transduced by adenovirus. In support of this, we found that sgRNA-mediated deletion of Blnc1 reached approximately 80% in fractionated hepatocytes (Supplementary Fig. 6b). In contrast, Blnc1 expression in the NPC fraction remained largely unaltered. As such, adenoviral sgRNA expression in Cas9 transgenic mice is an effective method to conditionally inactivate Blnc1 in the liver.

We next determined whether Blnc1 is causally linked to hepatic lipogenic activation in obesity. We transduced Cas9 transgenic mice with control (GFP) or Blnc1 sgRNA adenovirus via tail vein and subjected them to HFD feeding 3 weeks after transduction. Upon HFD feeding, Blnc1 LKO mice exhibited moderate resistance to diet-induced weight gain with approximately 15% lower body weight than control (Fig. 3b and Supplementary Fig. 7a). Fasting blood glucose and plasma insulin levels were significantly lower in Blnc1 LKO mice (Fig. 3c), suggesting that liver-specific inactivation of Blnc1 protects mice from HFD-induced insulin resistance. Accordingly, glucose tolerance test (GTT) and insulin tolerance test (ITT) revealed that HFD-fed Blnc1 LKO mice exhibited improved glucose tolerance and enhanced insulin sensitivity (Fig. 3d, e). Compared to control, plasma TAG level was lower in the LKO group, whereas total plasma cholesterol, non-esterified fatty acids (NEFA) and β-hydroxybutyrate concentrations remained similar between two groups (Fig. 3f).

We next examined how hepatocyte Blnc1 inactivation modulates the progression of HFD-induced hepatic steatosis. Liver mass and hepatic TAG and cholesterol content were significantly reduced in Blnc1 LKO mice following HFD feeding (Fig. 4a, b and Supplementary Fig. 7a). While BAT mass is slightly decreased, no significant difference in eWAT weight was observed between two groups (Supplementary Fig. 7b). Consistent with an important role of Blnc1 as an upstream activator of the SREBP1 pathway, protein levels of the precursor and nuclear isoforms of SREBP1 were greatly reduced by Blnc1 inactivation (Fig. 4c). Analysis of hepatic gene expression indicated that mRNA levels of key genes involved in lipogenesis, including *Srebp1c*, *Ppary*, *Gck*, *Fasn*, *Scd1*, *Fabp4*, *Dgat2*, *Fsp27*, and *Apoa4*, were significantly decreased (Fig. 4d). Blnc1 deficiency also attenuated the expression of genes involved in inflammatory response and cytokine/chemokine signaling, including *Adgre1*, *Ccl2*, *Ccr2*, *Cx3cr1*, *Lnc2*, and *Mmp12*. To rule out the possibility that Blnc1 sgRNA expression may affect hepatic lipogenesis, we transduced WT mice with GFP or sgRNA adenovirus and subjected them to HFD feeding. We did not observe significant effects of sgRNA expression on diet-induced weight gain and insulin resistance (Supplementary Fig. 7c–e).

Previous studies have demonstrated that hepatic lipogenesis is closely linked to adiposity and systemic metabolic physiology. Liver-specific transgenic expression of the transcriptionally active form of SREBP1c exacerbates hepatic steatosis and diet-induced weight gain[30,31], whereas inhibition of this pathway elicits beneficial metabolic effects[32,33]. We next performed metabolic cage studies to determine how hepatic Blnc1 inactivation influences whole body energy metabolism. Blnc1 LKO mice exhibited elevated $VO_2$ and energy expenditure rate that were accompanied by increased food intake and locomotor activity (Fig. 5a, b). Consistently, brown fat from Blnc1 LKO mice contained less fat and exhibited increased thermogenic gene expression (Fig. 5c, d). It is possible that hepatic Blnc1 inactivation may alter the secretion of liver-derived endocrine factors that regulate brown fat function; however, mRNA expression of the hepatokines Fgf21 and Gdf15 remained largely unaltered (Supplementary Fig. 7f). Together, these results demonstrate that Blnc1 is required for obesity-linked lipogenic activation and the development of HFD-induced insulin resistance and hepatic steatosis.

**Hepatic Blnc1 deficiency ameliorates NASH pathogenesis.** The observations that Blnc1 exerts powerful effects on hepatic lipid metabolism and inflammatory response raised an intriguing possibility that Blnc1 may be an important regulator of NASH pathogenesis. In addition to hepatic steatosis, the hallmarks of NASH include liver injury, chronic inflammation, and fibrosis. We recently demonstrated that a diet containing fructose, trans fats, and cholesterol was highly effective in recapitulating key features of human NASH in mice[9]. A remarkable aspect of this model is that NASH progression occurs in the context of obesity and insulin resistance, which more closely resembles human

**Fig. 2** Blnc1 is required for LXR-mediated lipogenic activation. **a** Plasma and liver TAG content from WT or Blnc1 KO mice receiving oral gavage of vehicle (Veh, open, WT, $n = 4$; KO, $n = 3$) or T0901317 (T1317, filled, WT, $n = 3$; KO, $n = 4$) for 4 days. **b** H&E staining of liver sections. Scale bar = 100 μm. **c** qPCR analysis of hepatic gene expression. Data in (**a**) and (**c**) represent mean ± SEM. *$p < 0.05$, **$p < 0.01$, ***$p < 0.001$, WT vs. KO, two-way ANOVA. **d** Immunoblots of total liver lysates and liver nuclear extracts. **e** qPCR analysis of gene expression in WT and Blnc1 KO hepatocytes treated with vehicle (Veh) or 5 μM T1317 for 24 h. **f** Incorporation of $^{14}C$-acetate into lipids in hepatocytes treated with Veh or T1317. Data in (**e**, **f**) represent mean ± SD ($n = 3$). *$p < 0.05$, **$p < 0.01$, ***$p < 0.001$, WT vs. KO, two-way ANOVA. **g** qPCR analysis of gene expression in hepatocytes transduced with GFP or Blnc1 adenovirus treated with Veh or 10 μM SR9238 for 24 h. **h** Incorporation of $^{14}C$-acetate into lipids in treated hepatocytes. Data in (**g**) and (**h**) represent mean ± SD ($n = 3$). **$p < 0.01$, ***$p < 0.001$, GFP vs. Blnc1, #$p < 0.001$, Veh vs. SR9238, two-way ANOVA

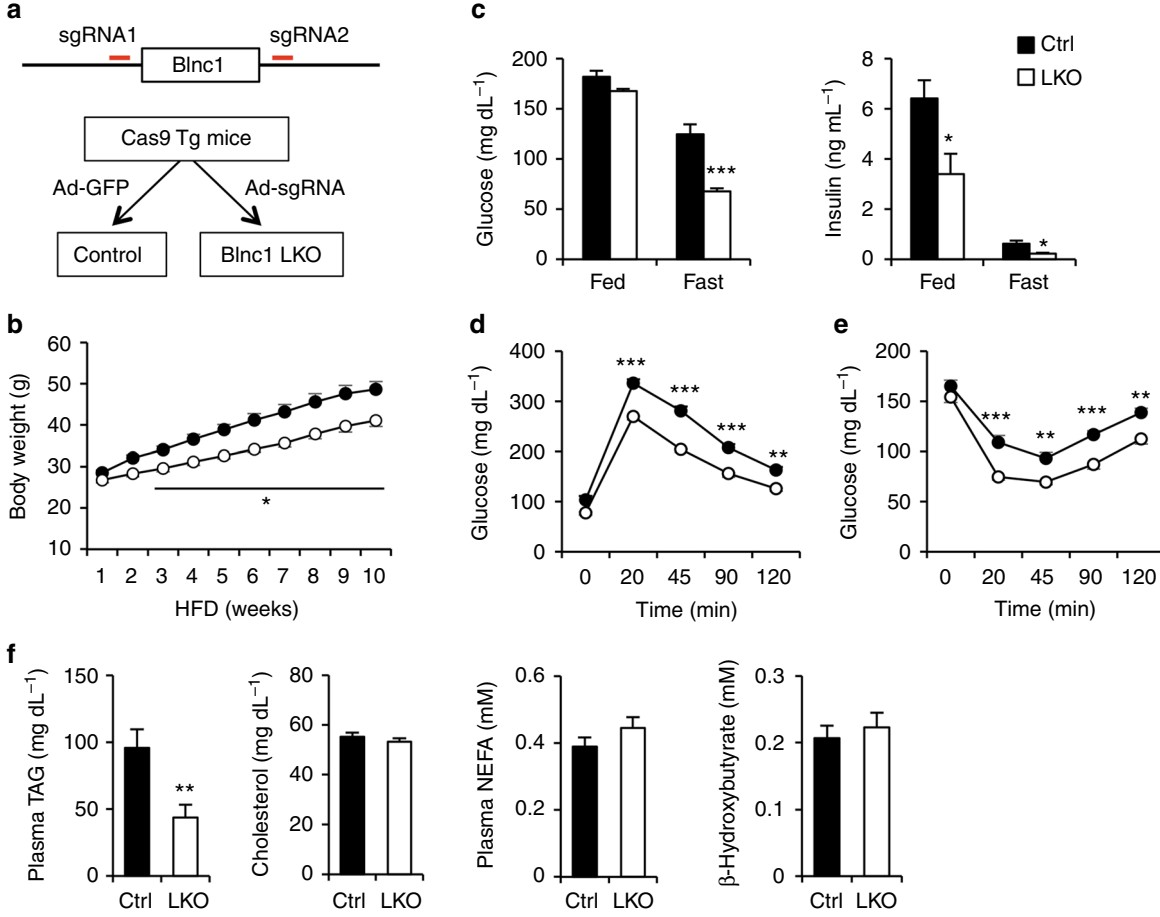

**Fig. 3** Liver-specific inactivation of Blnc1 alleviates diet-induced obesity. **a** A schematic of the strategy for the generation of Blnc1 LKO mice using CRISPR/Cas9. Adenoviral expression of sgRNAs in Cas9 transgenic mice results in liver-specific deletion of Blnc1. **b** Body weight of control (ctrl, filled, $n = 8$) and LKO (open, $n = 11$) mice fed HFD for 10 weeks. Data represent mean ± SEM. *$p < 0.01$, ctrl vs. LKO, two-way ANOVA with multiple comparisons. **c** Blood glucose (left) and plasma insulin (right) concentrations in ctrl ($n = 9$) and LKO ($n = 9$) mice after HFD feeding for 17 weeks (fast) and 19 weeks (fed). **d**, **e** GTT (**d**) and ITT (**e**) in ctrl (filled, $n = 9$) and LKO (open, $n = 9$) mice fed HFD for 8 and 9 weeks, respectively. Data represent mean ± SEM. **$p < 0.01$, ***$p < 0.001$; ctrl vs. LKO, two-way ANOVA with multiple comparisons. **f** Plasma parameters in mice fed HFD for 19 weeks. Data in (**c**) and (**f**) represent mean ± SEM. *$p < 0.05$, **$p < 0.01$, ***$p < 0.001$; ctrl vs. LKO, two-tailed unpaired Student's $t$-test

NASH compared to the commonly used methionine/choline deficient diet. qPCR analysis indicated that hepatic Blnc1 expression was elevated in mice following NASH diet feeding (Fig. 6a). To determine whether hepatic Blnc1 modulates diet-induced NASH, we fed control and Blnc1 LKO mice NASH diet for 24 weeks and monitored NASH parameters. Consistent with HFD feeding, we observed that liver mass and plasma and liver TAG and cholesterol levels were significantly lower in Blnc1 LKO mice (Fig. 6b, c). The mice also had reduced total plasma cholesterol. Measurements of plasma alanine aminotransferase (ALT) and aspartate aminotransferase (AST) levels revealed that Blnc1 LKO mice had approximately 39% and 33% reduction of these liver injury markers, respectively, compared to control (Fig. 6d). Further, Sirius red staining of liver sections indicated that Blnc1 LKO mice exhibited markedly improved fibrosis with a corresponding reduction of liver hydroxyproline content by approximately 39% (Fig. 6e, f). TUNEL staining revealed that the presence of apoptotic cells was significantly reduced as a result of Blnc1 inactivation (Fig. 6g).

To interrogate the global effects of Blnc1 deficiency on the liver transcriptome, we performed RNA sequencing analysis and identified a list of genes that were upregulated or downregulated following Blnc1 inactivation (Fig. 7a and Supplementary Table 1). Gene ontology analysis indicated that the downregulated genes

were enriched for collagen fibril organization, angiogenesis, cell adhesion, lipogenesis, and cell death, pathways linked to liver injury and fibrosis (Fig. 7b). In further support of this, qPCR analysis indicated that mRNA expression of genes involved in lipogenesis, liver fibrosis, and inflammation was significantly lower in mice with liver-specific Blnc1 deletion (Fig. 7c–e). The upregulated genes were enriched for epoxygenase p450 pathway, oxidation and reduction processes, and lipid and xenobiotic metabolism (Fig. 7b), suggesting that Blnc1 deficiency may improve hepatic detoxification and conversion of different lipid species. Compared to control, protein levels of the precursor and processed SREBP1 isoforms were decreased in Blnc1 LKO mouse livers (Fig. 7d). Further, JNK1/2 phosphorylation and cell death markers (PARP and CASPASE 3 cleavage) were attenuated in Blnc1 LKO mouse livers. Together, these data illustrated an important function of Blnc1 in the pathogenesis of diet-induced NASH progression.

**Blnc1 orchestrates the assembly of a lipogenic ribonucleoprotein transcriptional complex.** LncRNAs are known to form ribonucleoprotein complexes to exert their biological effects. We previously reported that Blnc1 physically interacts with the transcription factors early B-cell factor 2 (EBF2) and zinc finger

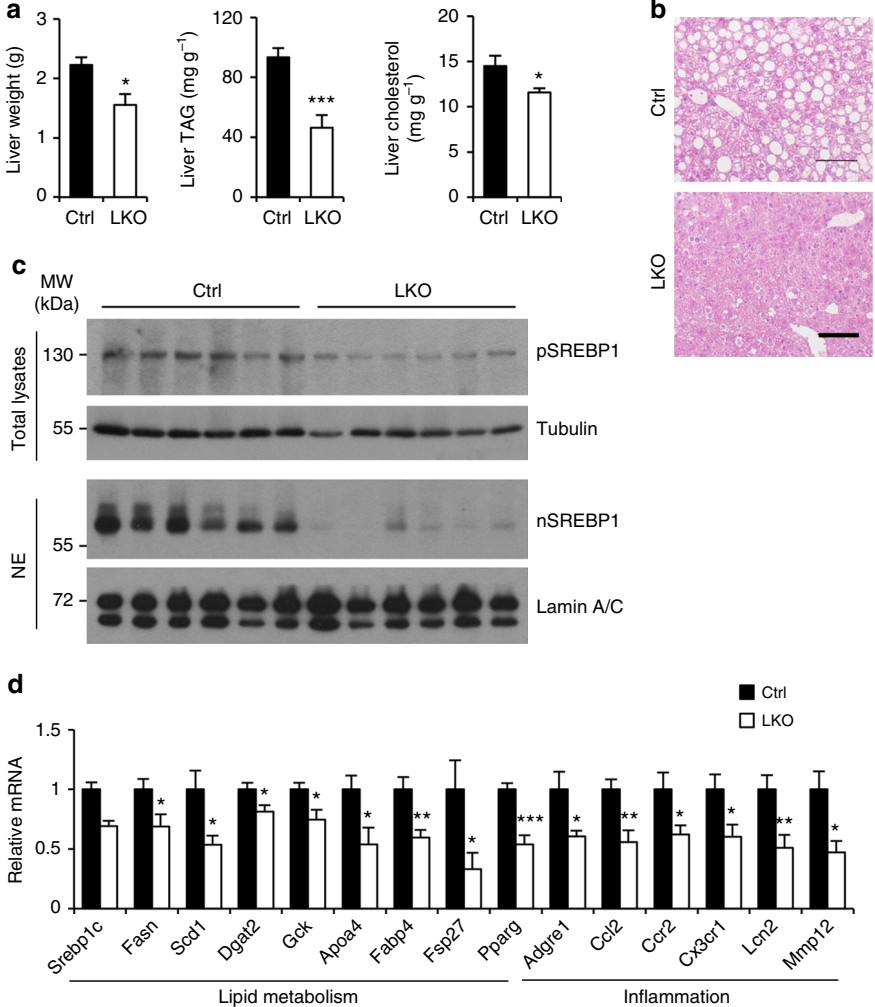

**Fig. 4** Liver-specific inactivation of Blnc1 attenuates hepatic lipogenesis and preserves metabolic health. **a** Liver weight, TAG, and cholesterol content in ctrl (filled, $n = 9$) and LKO (open, $n = 9$) mice fed HFD for 19 weeks. **b** H&E staining of liver sections. Scale bar = 100 μm. **c** Immunoblots of total liver lysates and liver nuclear extracts. **d** qPCR analysis of hepatic gene expression. Data in (**a**) and (**d**) represent mean ± SEM. *$p < 0.05$, **$p < 0.01$, ***$p < 0.001$; ctrl vs. LKO, two-tailed unpaired Student's $t$-test

and BTB domain-containing 7b (ZBTB7B) to promote thermogenic gene expression in brown and beige adipocytes[20,34]. The molecular nature of the Blnc1 ribonucleoprotein complex in lipogenic gene induction has not been defined. We next took a proteomic approach to identify protein factors that physically associate with Blnc1 in the liver. We constructed a recombinant AAV expressing Streptavidin Aptamer-tagged Blnc1 (StA-Blnc1) under the liver-specific TBG promoter. We isolated liver nuclei from mice transduced with control (AAV-GFP) or AAV-StA-Blnc1 and affinity-purified Blnc1-associated proteins using streptavidin agarose beads followed by mass spectrometry analysis. We identified Y-box Binding Protein 1 (YBX1) and endothelial differentiation-related factor 1 (EDF1) as two specific Blnc1-interacting proteins (Fig. 8a). The interaction between StA-Blnc1 and YBX1 and EDF1 was confirmed by co-immunoprecipitation (co-IP) analysis of endogenous proteins in the liver (Fig. 8b). Further, IP/qPCR studies indicated that the immunocomplexes of endogenous YBX1 and EDF1 in the liver were enriched for Blnc1 RNA (Fig. 8c). We also observed that Blnc1 physically associates with heterogeneous nuclear ribonucleoprotein U (hnRNPU), a protein known to interact with Blnc1[23], suggesting that hnRNPU is likely a core component of the Blnc1 ribonucleoprotein complex.

EDF1 is an RNA-binding protein that was previously reported to interact with nuclear hormone receptors, including LXRα[35,36]. We next examined the physical and functional relationships among LXRα, EDF1, and Blnc1. We transiently transfected HEK293T cells with plasmids expressing Blnc1 and HA-tagged EDF1 or LXRα alone or in combination and performed IP/qPCR assay using α-HA agarose beads. This protein-RNA interaction assay indicated that Blnc1 forms a physical complex with EDF1 and LXRα (Fig. 8d, e). In contrast, we did not observe interaction between Blnc1 and retinoid X receptor β (RXRβ) (Supplementary Fig. 8a). The association between Blnc1 and LXRα appeared to be ligand-independent. Conversely, we detected EDF1 and LXRα in Blnc1 ribonucleoprotein complexes following streptavidin beads precipitation of lysates from transiently transfected HEK293T cells (Fig. 8f). These protein-RNA interaction studies demonstrate that Blnc1 forms a ribonucleoprotein transcriptional complex with EDF1 and LXR.

**Blnc1 facilitates the formation of LXR/EDF1 complex and enhances EDF1 activity.** We next dissected the RNA domains (RD) of Blnc1 that mediate its interaction with LXRα and EDF1 (Fig. 9a). We previously demonstrated that RD1 is required for its

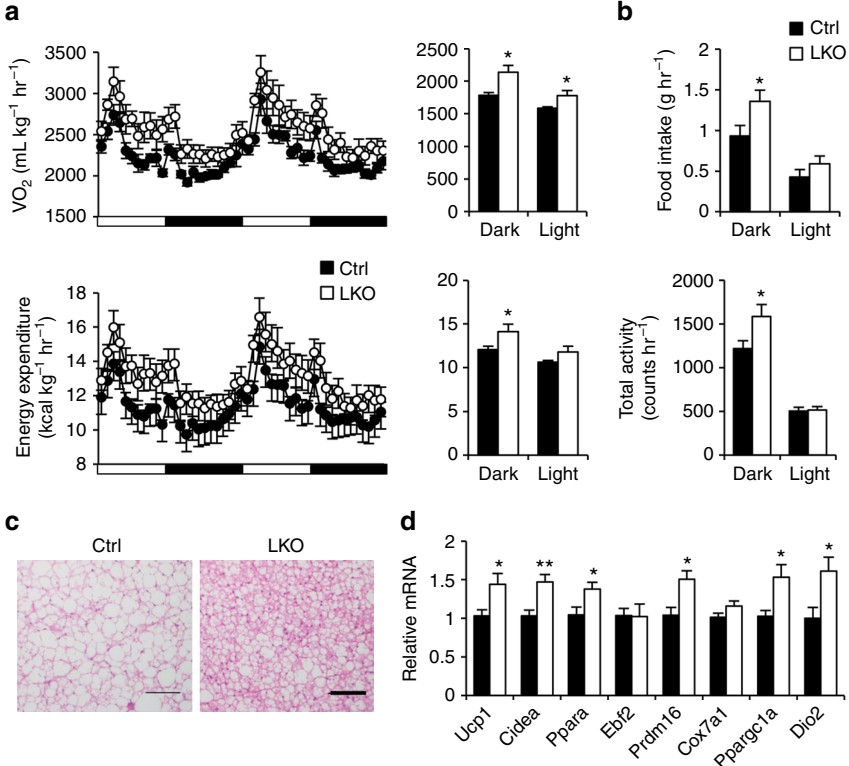

**Fig. 5** Hepatic ablation of Blnc1 stimulates thermogenesis and energy expenditure. **a, b** VO₂ and energy expenditure (**a**), food intake and total activity (**b**) in ctrl (filled, $n = 8$) and LKO (open, $n = 8$) mice fed HFD for 15 weeks. Data in (**a**) and (**b**) represent mean ± SEM. *$p < 0.05$; ctrl vs. LKO, two-tailed unpaired Student's $t$-test. **c** H&E staining of BAT section from HFD-fed mice. Scale bar = 100 μm. **d** qPCR analysis of brown fat gene expression in HFD-fed ctrl (filled, $n = 9$) and LKO (open, $n = 9$) mice. Data represent mean ± SEM. *$p < 0.05$, **$p < 0.01$; ctrl vs. LKO, two-tailed unpaired Student's $t$-test

interaction with hnRNPU and adipogenic function[23]. Interestingly, deletion of this fragment did not impair interaction between Blnc1 and LXRα and EDF1 (Fig. 9b). In contrast, RD2 and RD3 appeared to be required for their physical association. To assess the possibility that EDF1 may facilitate the recruitment of Blnc1 to the LXR transcriptional complex, we performed RNA precipitation followed by immunoblotting in transiently transfected HEK293T cells. As shown in Fig. 9c, co-expression of EDF1 increased the recruitment of LXRα to the precipitated Blnc1 RNA complex. Consistently, IP/qPCR assay indicated that the recruitment of Blnc1 to LXRα was significantly increased by EDF1 co-expression (Fig. 9d).

EDF1 is known to serve as a transcriptional coactivator for multiple transcription factors[37,38]. It is possible that Blnc1 may facilitate the recruitment and coactivation of LXR by EDF1. To test this, we transfected HA-LXRα and Flag-EDF1 in HEK293T cells in the presence or absence of Blnc1. We detected physical association between LXRα and EDF1 that was further augmented by Blnc1 expression (Fig. 9e). Compared to control hepatocytes, physical interaction between LXRα and EDF1 was markedly reduced in hepatocytes lacking Blnc1 (Fig. 9f). Luciferase reporter assay indicated that EDF1 acted in concert with LXRα/RXRβ to stimulate *Srebp1c* promoter activity and that Blnc1 further enhanced the activity of this transcriptional complex (Fig. 9g). Further, chromatin immunoprecipitation studies revealed that the recruitment of LXRα to the LXR binding site on the proximal *Srebp1c* promoter was greatly impaired in the absence of Blnc1 (Fig. 9h). These results illustrate close physical and functional interactions between LXRα, EDF1, and Blnc1 in hepatocytes.

To establish the role of Blnc1 in mediating the effects of EDF1, we overexpressed EDF1 in primary hepatocytes using adenoviral

transduction. EDF1 overexpression in hepatocytes stimulated lipogenic gene expression under basal and LXR activating conditions. Co-expression of Blnc1 further enhanced the stimulatory effects of EDF1 on lipogenic gene expression (Fig. 10a). Importantly, EDF1-induced lipogenic gene expression was significantly blunted in hepatocytes lacking Blnc1 (Fig. 10b). Finally, we performed siRNA knockdown of EDF1 using two independent siRNA pools in primary hepatocytes transduced with GFP or Blnc1 adenovirus. qPCR analysis revealed that, compared to control, both siRNAs drastically reduced endogenous EDF1 expression (Supplementary Fig. 8b). Importantly, the induction of *Srebp1c* and *Fasn* expression by Blnc1 and LXR agonist was significantly blunted by siRNA knockdown of EDF1 (Fig. 10c), indicating that EDF1 is required for full lipogenic activation by LXR and Blnc1. Together, these results outline a molecular mechanism through which EDF1 facilitates the recruitment of Blnc1 and the formation of the LXR ribonucleoprotein transcriptional complex (Fig. 10d).

## Discussion

Aberrant activation of hepatic lipogenesis has been implicated in the pathogenesis of metabolic disorders in obesity, particularly NAFLD. However, the regulatory network that governs physiological and pathological activation of the lipogenic program remains incompletely understood. LncRNAs are emerging as an important new class of regulators that impinge on diverse biological processes and pathogenesis of the metabolic diseases[17,39]. These functional RNAs interface with protein factors through the formation of ribonucleoprotein complexes to alter chromatin structure and gene transcription. To date, only a small number of functional lncRNAs have been identified to regulate metabolic

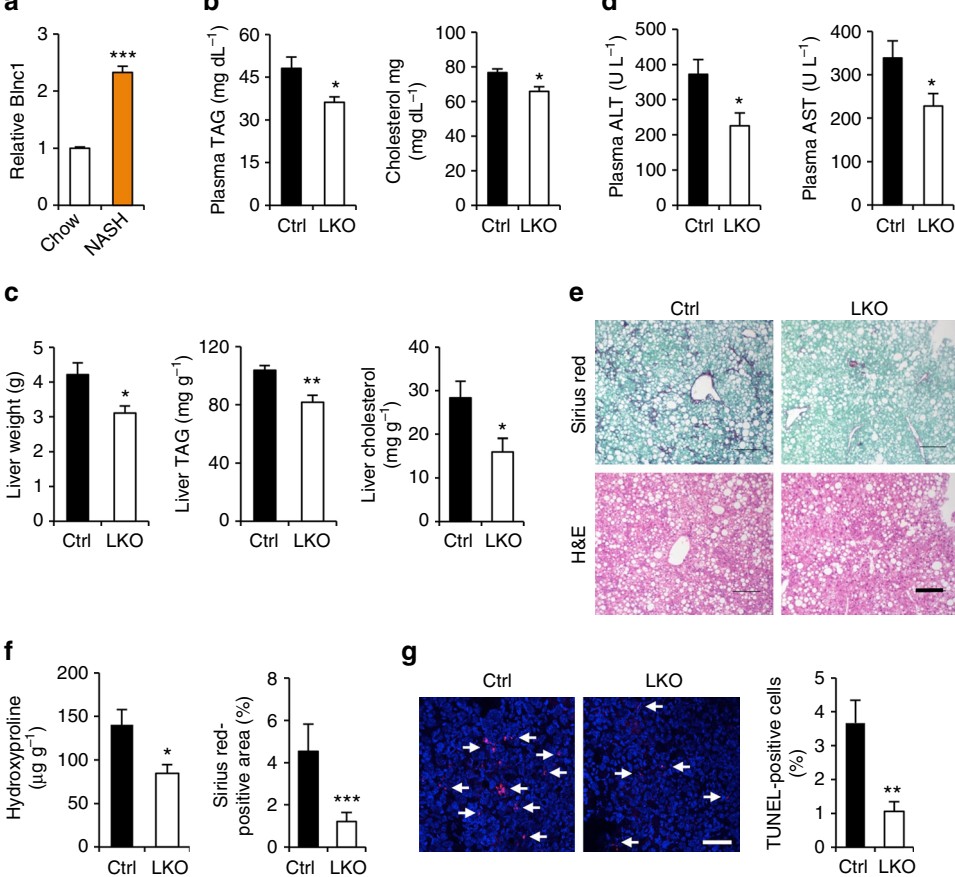

**Fig. 6** Blnc1 inactivation ameliorates diet-induced NASH. **a** Blnc1 expression in mice fed chow ($n = 3$) or NASH ($n = 3$) diet for 6 months. Data represent mean ± SEM. ***$p < 0.001$, chow vs. NASH, two-tailed unpaired Student's $t$-test. (**b**–**d**) Plasma lipids (**b**), liver weight, TAG and cholesterol content (**c**) and plasma ALT and AST levels (**d**) in ctrl (filled, $n = 6$) and LKO (open, $n = 6$) mice fed NASH diet for 24 weeks. **e** H&E and Sirius red staining of liver sections. Scare bar = 100 μm. **f** Liver hydroxyproline content and quantitation of Sirius red-positive area in **e**. **g** TUNEL staining and quantitation (scale bar = 100 μm). Data in (**b**–**d**), (**f**) and (**g**) represent mean ± SEM. *$p < 0.05$, **$p < 0.01$, ***$p < 0.001$. Ctrl vs. LKO, two-tailed unpaired Student's $t$-test

tissue development and metabolic physiology. Whether lncRNAs play a role in the control of hepatic lipogenesis and NAFLD pathogenesis remains largely unknown. In this study, we demonstrate that Blnc1 is a core component of the LXR/SREBP1c pathway that is required for lipogenic induction in the insulin resistant state.

Hepatic Blnc1 expression is strongly linked to activation of lipogenesis in mouse models of obesity and NASH. While the exact signal that triggers Blnc1 induction remains unknown, we observed that fatty acids and TNFα increased Blnc1 expression in cultured hepatocytes (data not shown). LXR activation provides a critical upstream signal that drives hepatic lipogenesis in response to feeding and during obesity[14,15]. In this context, Blnc1 augments the stimulatory effects of the LXR agonist T0901317 on de novo lipogenesis and is required for LXR-mediated lipogenic activation in cultured hepatocytes and in vivo. Mice with liver-specific inactivation of Blnc1 were resistant to HFD-induced hepatic steatosis and had lower plasma TAG levels, metabolic changes characteristic of diminished hepatic lipogenesis. Remarkably, hepatic Blnc1 deletion also protected mice from diet-induced obesity, insulin resistance, and hepatic inflammation. We further demonstrated that mice lacking Blnc1 in hepatocytes exhibited significantly reduced liver injury and fibrosis accompanied by attenuation of inflammation following NASH diet feeding. These observations are consistent with several previous studies that demonstrated a central role of hepatic lipogenic induction in driving metabolic disorders in obesity[30–32,40]. Our

findings support a "licensing" role of Blnc1 and its protein partners in obesity-linked activation of hepatic lipogenesis and NAFLD pathogenesis (Fig. 10d).

Previous work has been demonstrated that Blnc1 forms ribonucleoprotein transcriptional complexes with EBF2 and ZBTB7B in brown adipocytes[20,23,34]. However, the expression of these adipogenic transcriptional factors is low in hepatocytes, suggesting that Blnc1 may engage other factors in lipogenic gene activation. Our functional and biochemical studies support a crucial role of Blnc1 in the lipogenic response mediated by the LXR-SREBP1c pathway. Proteomic analysis of the hepatic Blnc1 ribonucleoprotein complexes identified several Blnc1-interacting proteins, including EDF1, YBX1, and hnRNPU. Interestingly, distinct RNA domains appeared to mediate its interaction with protein factors. While the role of YBX1 and hnRNPU in lipogenic regulation remains currently unknown, EDF1 is emerging as an important factor that facilitates the assembly of the LXR transcriptional complex and lipogenic gene induction. A recent study identified LeXis as an LXR-regulated lncRNA that functions as a negative feedback mechanism for cholesterol biosynthesis[22]. As such, it is likely that the integration of lncRNA and protein regulators plays a much broader role in metabolic gene expression, systems physiology and disease pathogenesis.

## Methods
**Mice**. All animal studies were performed according to procedures approved by the Institutional Animal Care & Use Committee at the University of Michigan. Mice

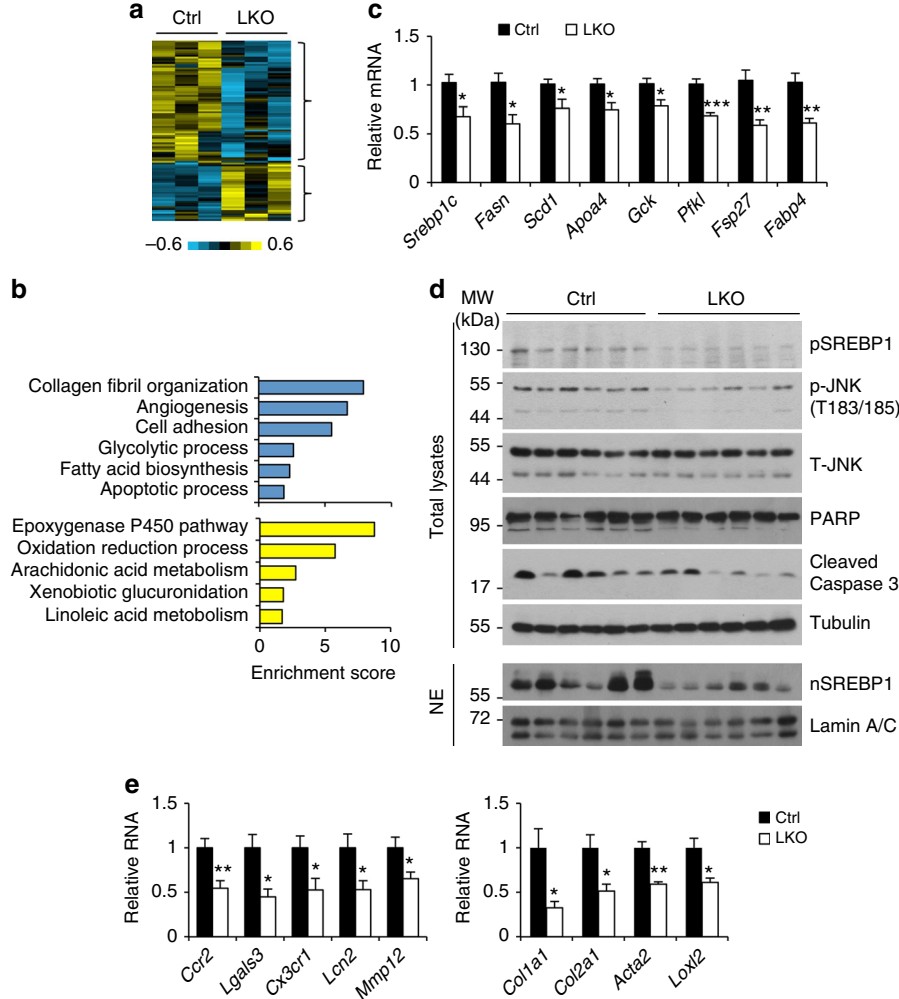

**Fig. 7** Blnc1 inactivation suppresses the molecular signature of diet-induced NASH. **a** Heat map representation of upregulated and downregulated genes identified in RNAseq analysis of the livers from NASH-fed mice. **b** Enrichment score for downregulated (blue) and upregulated (yellow) genes. **c** qPCR analysis of hepatic genes involved in lipid metabolism from ctrl (filled, $n = 6$) and LKO (open, $n = 6$) mice fed NASH diet for 24 weeks. **d** Immunoblots of total liver lysates and liver nuclear extracts from NASH-fed mice. **e** qPCR analysis of hepatic genes involved in inflammation and fibrosis. Data in (**c**) and (**e**) represent mean ± SEM. *$p < 0.05$, **$p < 0.01$, ***$p < 0.01$; ctrl vs. LKO, two-tailed unpaired Student's $t$-test

were maintained in 12/12 h light/dark cycles and fed standard rodent chow or HFD (D12492, Research Diets) or NASH diet (D09100301, Research Diets). WT C57BL/6J mice (JAX #000664) and Cas9 transgenic mice (JAX #024858) were purchased from the Jackson Laboratory. To generate Blnc1 single-guide RNA (sgRNA) transgenic mice, we designed two sgRNAs (sgRNA1: GGGTGGAACTG-TAAGGCGTG and sgRNA2: GCTTGCTAGATAGCTTGTGC) flanking the mouse Blnc1 gene (http://crispr.mit.edu/)[29]. To generate Blnc1 sgRNA transgenic mice, we constructed a tandem U6-sgRNA transgenic construct and purified the expression cassette for microinjection into fertilized eggs. The constitutive Cas9 transgenic mice were mated with Blnc1 sgRNA transgenic mice to generate germline Blnc1 deletion. The Cas9 and Blnc1 sgRNA transgenic cassettes were removed following backcrossing with WT C57BL/6J mice to generate germline Blnc1 heterozygous mice and later whole body Blnc1 KO mice. Genotyping primer sequences are: P1: CAACCCGCAGTTACTATTTGGTG; P2: CCCCAA-CATTATTAGAGTCCAAGG; P3: GCCTTCAAGTCCATGGCGTAT.

Liver-specific Blnc1 KO mice were generated by transducing Cas9 transgenic mice with a recombinant adenoviral vector expressing two sgRNAs targeting Blnc1. For adenoviral transduction, we injected approximately 0.15 OD of purified adenoviral particles per mouse via tail vein. For AAV transduction, we injected approximately $1 \times 10^{11}$ genome copies of AAV vectors per mouse via tail vein. For oral gavage experiment, mice were given a daily dose of vehicle or 25 mg/kg T0901317 dissolved in sunflower oil for 4 days. Metabolic cage measurements were performed at the University of Michigan Animal Phenotyping Core. Oxygen consumption (VO2), spontaneous motor activity and food intake were measured using the Comprehensive Laboratory Animal Monitoring System (CLAMS, Columbus Instruments) according to manufacturer's instruction. For histology, tissues were dissected and immediately fixed in 10% formalin at 4°C overnight and processed for paraffin embedding and H&E staining. Sirius red staining was

performed as previously described[41]. Percentage of Sirius red staining positive area of the total area of view was quantified using ImageJ. For mouse studies, we randomly assigned mice of different genotypes to control and treatment groups. Measurements were performed without the knowledge of mouse genotypes and treatments. All mouse experiments were independently replicated for at least twice. We did not exclude any data points or mice unless a technical issue or human error occurs.

**Cell culture**. Primary hepatocytes were isolated from C57BL/6J mice, as previously described[42]. Hepatocytes were maintained in DMEM supplemented with 10% bovine growth serum (BGS) at 37°C and 5% $CO_2$. Adenoviral transduction was performed at the same day of hepatocyte isolation. Transduced hepatocytes were treated with vehicle (DMSO) or T0901317 (5 μM) for 24 h before analysis. HEK293T cells were purchased from American Type Culture Collection (ATCC) and cultured in DMEM supplemented with 10% FBS. To knock down Edf1 in primary hepatocytes, two independent siRNAs targeting *Edf1* (Thermo Fisher, ID: S81801 and S81802) and control siRNA were transfected using Lipofectamine RNAiMAX Transfection Reagent (Thermo Fisher). The cell culture experiments were performed in triplicates and repeated at least three times.

**Metabolic analyses**. For glucose tolerant test (GTT), mice were fasted overnight (16 h) and i.p. injected with a glucose solution at 1.0 g/kg body weight. For ITT, mice were fasted for 4 h before i.p. injection of insulin at the dose of 1 U/kg body weight. Blood glucose concentrations were measured before and 20, 45, 90, and 120 min after glucose or insulin injection. Liver TAG was extracted and measured as previously described[43]. Plasma insulin was measured using an ELISA kit (CrystalChem). Plasma TAG, NEFA, β-hydroxybutyrate, ALT, AST, and

 

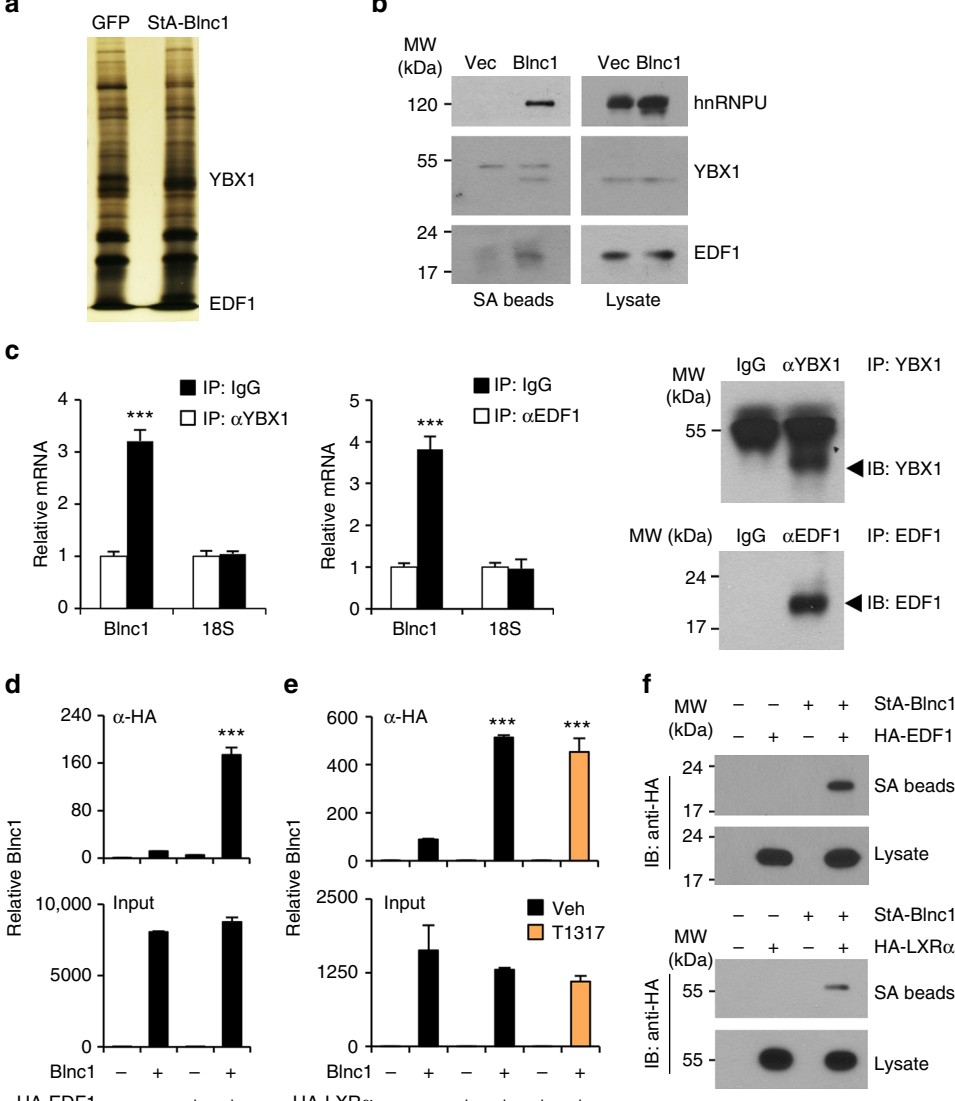

**Fig. 8** Proteomic analysis of the liver Blnc1 ribonucleoprotein complex. **a** Silver-stained gel of affinity-purified Blnc1 complexes. **b** Immunoblots of precipitated proteins on streptavidin (SA) beads or total liver lysates. **c** IP/qPCR analyses of endogenous Blnc1 on the immunocomplexes as indicated. Immunoblots of precipitated proteins are shown on the right. Data represent mean ± SD ($n = 3$). ***$p < 0.001$, two-tailed unpaired Student's $t$-test. **d** IP/qPCR analyses of Blnc1 in EDF1 (**d**) or LXRα (**e**) immunocomplexes (top) or input (bottom) from transiently transfected HEK293T cells. Data in (**d**) and (**e**) represent mean ± SD ($n = 3$). ***$p < 0.001$, two-way ANOVA. **f** Immunoblots of HA-EDF1 (top) or HA-LXRα (bottom) from total lysates and precipitated proteins on SA beads from transfected HEK293T cells

cholesterol were measured using commercial kits. Liver hydroxyproline content was measured using Hydroxyproline Colorimetric Assay Kit (BioVision). TUNEL staining was performed on frozen liver sections using an ApoBrdU DNA Fragmentation Assay Kit (BioVision). Percentage of TUNEL-positive cells among DAPI-stained cells was quantified using ImageJ, as previously described[9].

**Gene expression analyses**. Mouse liver and fat tissue were extracted, immediately frozen in liquid nitrogen. Total RNA from tissues or hepatocytes was extracted using TRIzol method following manufacturer instructions. For RT-qPCR, 2 μg of total RNA was reverse-transcribed using MMLV (Invitrogen) followed by qPCR using SYBR Green (Life Technologies). Relative mRNA expression was normalized to internal control ribosomal protein 36B4. The qPCR primers used for gene expression are listed in Supplementary Table 2.

**Immunoblotting analysis**. Total liver lysates were prepared by homogenizing in a buffer containing 50 mM Tris (pH = 7.6), 130 mM NaCl, 5 mM NaF, 25 mM β-glycerophosphoate, 1 mM sodium orthovanadate, 10% glycerol, 1% Triton X-100, 1 mM DTT, 1 mM PMSF, and the protease inhibitor cocktail (Roche). Total cell lysates were prepared in a lysis buffer containing 50 mM Tris-HCl (pH = 7.8), 137

mM NaCl, 10 mM NaF, 1 mM EDTA, 1% Triton X-100, 10% glycerol, and the protease inhibitor cocktail after three freeze/thaw cycles. Liver nuclear extracts were prepared using a Dounce homogenizer in a buffer containing 0.6% NP40, 150 mM NaCl, 10 mM HEPES (pH = 7.9), 1 mM EDTA, and protease inhibitor cocktail. Tissue debris was removed by a brief centrifugation at 500 r.p.m. 4 °C. Nuclei were collected from supernatant by centrifugation at 3000 r.p.m. for 5 min and resuspended in a lysis buffer containing 4% SDS, 20% glycerol, 100 mM dithiothreitol and Tris-HCl (pH = 6.8). The antibodies used are: anti-hnRNPU (sc-32315), anti-SREBP1 (sc-13551), and anti-HA (sc-805) from Santa Cruz Biotechnology; anti-EDF1 (12419-1-AP, Proteintech); anti-LXRα (ab41902, Abcam); anti-Tubulin (T6199, Sigma); anti-phospho-JNK1/2 (T183/Y185) (4668), anti-total-JNK1/2 (9252), anti-PARP (9542), anti-cleaved Caspase3 (9661) and anti-YBX1 (4202) from Cell Signaling Technology. All the antibodies were used at a 1:1000 dilution. Uncropped blots are shown in Supplementary Figs. 9–11.

**RNA-seq data analysis**. RNA sequencing was performed on polyA-selected mRNA from NASH diet-fed control and Blnc1 LKO mice. The Fastq files were aligned to the mouse reference genome (mm10) using the aligner STAR[44]. HTSeq was used to count reads mapped to each gene feature[45]. DESeq2 was used for

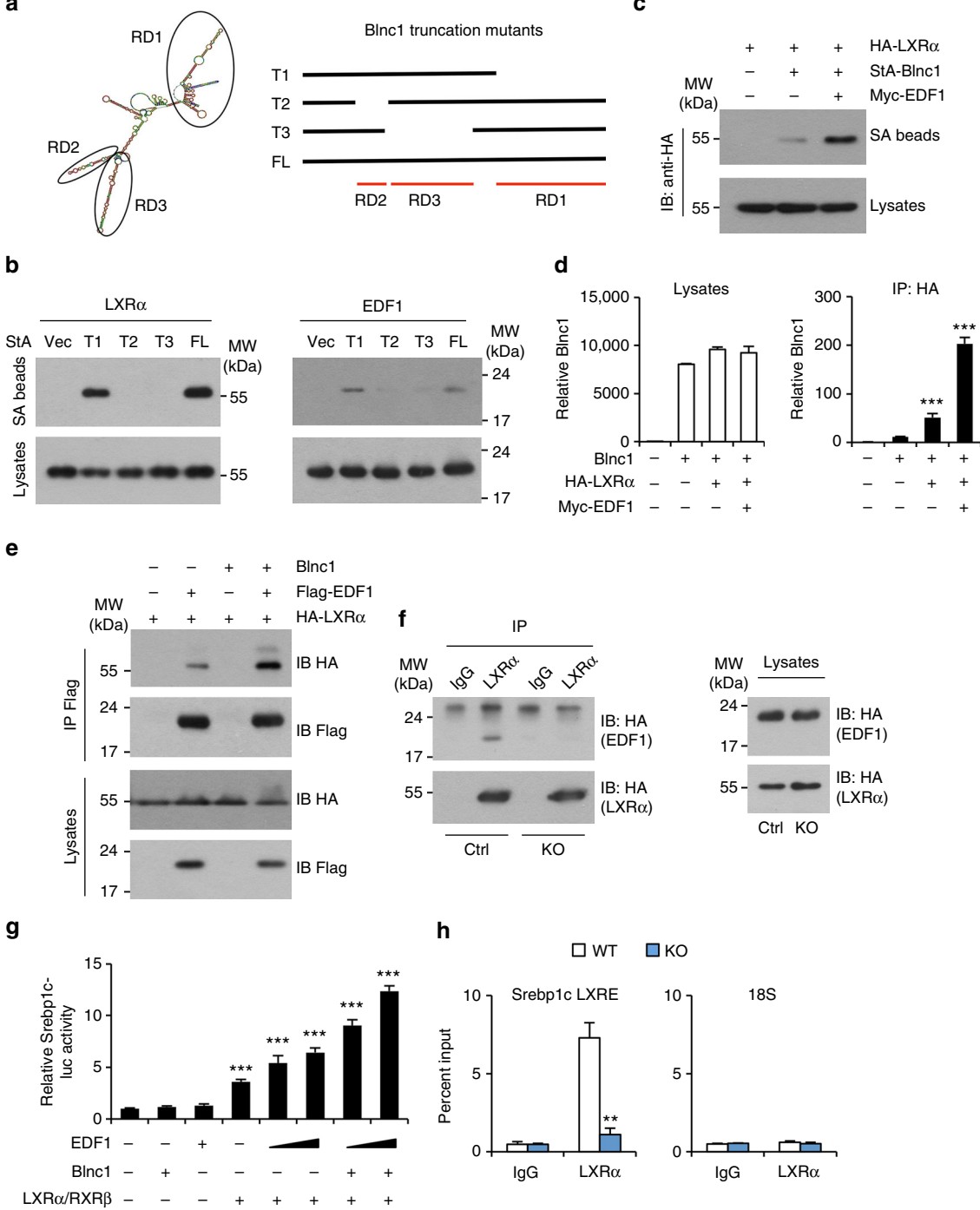

**Fig. 9** Physical and functional interaction between Blnc1, LXRα, and EDF1. **a** Diagram of Blnc1 RNA domains (RD, left) and truncation mutants (right). **b**, **c** Immunoblots of total lysates and streptavidin (SA) beads precipitated proteins from transiently transfected HEK293T cells. **d** IP/qPCR analyses of Blnc1 in α-HA immunocomplexes (right) and input (left) from transiently transfected HEK293T cells. **e** Immunoblots of total lysates and α-Flag immunocomplexes from transiently transfected HEK293T cells. **f** Immunoblots of total lysates and α-LXRα immunocomplexes from WT and Blnc1 KO hepatocyte transduced with HA-LXRα and HA-EDF1 adenoviruses. **g** *Srebp1c*-luciferase reporter assay. Data in (**d**) and (**g**) represent mean ± SD ($n = 3$). ***$p < 0.001$, vs. vector, one-way ANOVA. **h** ChIP assay of LXRα occupancy in the livers from WT (open) and KO (filled) mice gavaged with T1317 for 4 days. Data represent mean ± SD ($n = 3$). **$p < 0.01$, two-tailed unpaired Student's *t*-test

differential expression analysis[46]. Gene enrichment analysis was performed using DAVID (david.abcc.ncifcrf.gov).

**In vitro lipogenesis assay.** Primary hepatocytes were transduced with recombinant adenoviruses for 24 h followed by treatment with DMSO or T0901317 for additional 24 h. For lipogenic assay, treated hepatocytes were incubated with Krebs Ringer buffer supplemented with 0.1% fatty acid free-BSA in the presence of 0.2 μCi/mL $C^{14}$-acetate and incubated for 2 h. Total lipids were extracted from treated

hepatocytes using a hexane-isopropanol mixture (hexane: isopropanol (v: v) = 3:2) and resuspended in toluene following solvent evaporation. Radioactivity was quantitatively measured using a scintillation counter.

**RNA-protein interaction assays.** RNA-protein interaction studies were performed as previously described[20]. Briefly, for IP/qPCR assay, HEK293T cells transiently transfected with HA-EDF1 or HA-LXRα and Blnc1 alone or in combinations. Immunoprecipitation was performed using anti-HA agarose beads. After

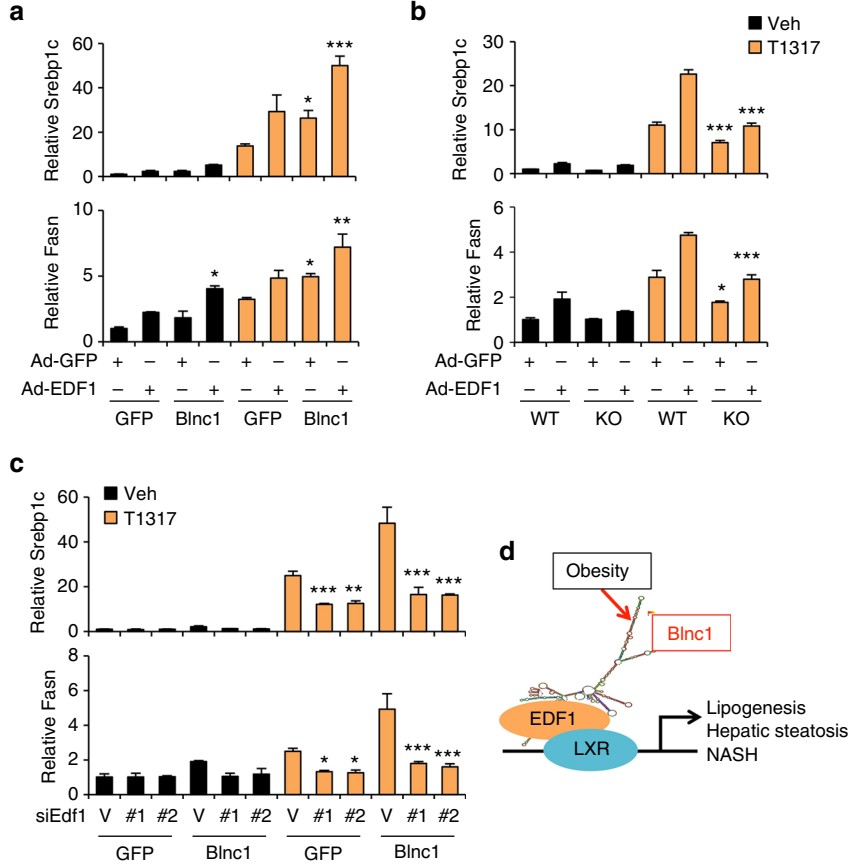

**Fig. 10** Role of EDF1 in lipogenic gene regulation. **a** qPCR gene expression analyses in hepatocytes transduced with recombinant adenoviruses as indicated. Data represent mean ± SD ($n = 3$). *$p < 0.05$, **$p < 0.01$, ***$p < 0.001$, GFP vs. Blnc1, two-way ANOVA. **b** Gene expression analysis in transduced WT or Blnc1 KO hepatocytes. Data represent mean ± SD ($n = 3$). *$p < 0.05$, ***$p < 0.001$, WT vs. KO, two-way ANOVA. **c** qPCR analysis of gene expression in transduced hepatocytes receiving control siRNA (V) or siRNAs targeting EDF1 (#1 and #2) followed by treatments with vehicle or 5 μM T1317 for 24 h. Data represent mean ± SD ($n = 3$). *$p < 0.05$, **$p < 0.01$, ***$p < 0.001$, vector vs. #1 or #2, two-way ANOVA. **d** A model depicting licensing of hepatic lipogenesis and NAFLD progression by the LXR/EDF1/Blnc1 ribonucleoprotein transcriptional complex

washing, RNA was extracted with Trizol, treated with RNase-free DNase, and analyzed by RT-qPCR. For reverse precipitation, StA-Blnc1 was precipitated from total lysates of transfected HEK293T cells using streptavidin agarose beads. Proteins associated with beads were analyzed by immunoblotting. For mass spectrometry analysis, WT mice were transduced with GFP or StA-Blnc1 AAV virus. Liver nuclear extract was prepared for affinity purification of Blnc1-associated proteins using streptavidin agarose beads. Protein bands unique for StA-Blnc1 were excised for identification by mass spectrometry. For RNA IP assay, liver nuclear extracts were incubated with control IgG or antibodies specific for YBX1 and EDF1 in the presence of protein A beads. Precipitated RNA was extracted using Trizol after three washes followed by RNase-free DNase treatment and subjected for RT-qPCR analysis.

**Luciferase reporter and ChIP assays**. The *Srebp1c* promoter was amplified by PCR and cloned into pGL3-Basic vector. HEK293T cells in a 12-well plate were transiently transfected with *Srebp1c*-luc (50 ng/well) and LXRα (100 ng/well), RXRβ (100 ng/well) in combination with Blnc1 (100 ng/well), and two different doses of EDF1 (100 and 200 ng/well). Cells were harvested for luciferase assay 36 h after transient transfection. All reporter assays were repeated at least three times in triplicates. For ChIP assay, chromatin extracts were prepared from WT and Blnc1 KO liver after T0901317 treatment for 4 days followed by preclearing and immunoprecipitation using control IgG or LXRα antibody (Abcam). Chromatin association was measured by qPCR using primers listed in Supplementary Table S2.

**Statistical analysis**. Statistical analysis was performed using GraphPad Prism 7. Statistical differences were evaluated using two-tailed unpaired Student's *t*-test for comparisons between two groups, or analysis of variance (ANOVA) and appropriate post hoc analyses for comparisons of more than two groups. Two-way ANOVA with multiple comparisons was used for statistical analysis of ITT, GTT, and body weight studies. A *p* value of less than 0.05 (*$p < 0.05$, **$p < 0.01$, and

***$p < 0.001$) was considered statistically significant. Statistical methods and corresponding *p* values for data shown in each panel were indicated in figure legends.

**Data availability**. Blnc1 sequence has been deposited into GenBank with accession number MG736818. RNA-seq data files have been deposited into the Gene Expression Omnibus database (www.ncbi.nlm.nih.gov/geo/) with accession number GSE108609.

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

## Acknowledgements

This work was supported by NIH (DK102456 and DK084771 J.D.L.), American Diabetes Association (1-15-BS-118, J.D.L.). X.Y.Z. is supported by NIH Pathway to Independence Award (DK106664). This work used core services supported by the Michigan Diabetes Research Center and the Michigan Nutrition and Obesity Research Center (DK020572 and DK089503).

## Author contributions

J.D.L. and X.Y.Z. conceived the project and designed research. X.Y.Z., X.X., T.L., L.M., X.P., C.R., L.G., S.L., and X.L. performed the studies. X.Y.Z. and J.D.L. analyzed the data and wrote the manuscript.

## Additional information

**Competing interests:** The authors declare no competing interests.

