## [Peer Review File · Nature Communications]

Reviewers' comments:

Reviewer #1 (expert in lipid metabolism and nuclear receptors)(Remarks to the Author):

The manuscript entitled "Long noncoding RNA licensing of obesity-linked hepatic lipogenesis and NAFLD pathogenesis" by Zhao et al is announced as an original work contribution. In this work the authors expand the role of the metabolic regulator lncRNA blnc1 showing that it contributes to hepatic lipogenesis and NAFLD by acting in concert with LXR and other transcriptional coactivators to induce Srebp1c expression.

The manuscript is well written and the figures are nicely rendered. Although the range of blnc1 genome-wide interactions are not well defined, on balance the work is novel, timely, and will provide an important impetus to examining mechanisms by which ligand-dependent transcription factors influence gene expression.

A few issues that need to be addressed

1. The LXR/Srebp1c axis is critical for hepatic responses in fasting/refeeding. Is there evidence that blnc1 is regulated during fasting and refeeding? Does loss of hepatic blnc1 impart a deficiency in hepatic responses to refeeding? Is the administration of insulin to blnc1 knockout hepatocytes sufficient to induce Srebp1c? These studies will strongly reinforce the main premise of the paper in a highly relevant physiologic context.

2. LXR activation has been shown to influence a large number of genes involved in hepatic metabolism including Abcg5, Lpcat3, rnf145, Abcg8 etc...Do the authors observe changes in canonical transcriptional targets of LXR as a consequence of loss of function and overexpression of blnc1? If the effect is restricted only to Srebp1c the authors should address this in the discussion. For example, could it be that blnc1 is deployed to only a subset of genes, are there unique coactivators that may be present at different sites, or perhaps LXR isoform specific interactions?

3. Very little information is provided on the characterization of the blnc1 transcript itself in liver. What is the copy number of blnc1 in hepatocytes or liver? In addition the authors should provide single molecule RNA FISH evidence that overexpressed biotin tagged blnc1 is localized to nucleus and ideally show that it colocalizes at Srebp1c genomic locus.

Other issues

4. Protein blots of nuclear and membrane form for Srebp1c and Srebp2 isoform should be provided.

5. Please provide cholesterol levels in serum and liver for data in figures 1 and 2.

6. One of the strengths of this work is use of novel approaches to generating various knockout models which would be of interest to the wider scientific community however detailed methods are not provided. For example specify name of vector used and precise cloning strategy as well as genetic background of all mice. Please provide these additional details in methods.

7. Similarly, the source data is not described for human tissue panel in supplementary figure 1a as well as data in figure 1b.

Reviewer #2 (expert in lncRNAs)(Remarks to the Author):

In this interesting manuscript, Zhao and colleagues report that the long noncoding RNA Blnc1 is part of the LXR transcriptional complex required for SREBP1c induction and hepatic lipogenic activation in obesity. Importantly, liver-specific ablation of Blnc1 prevented hepatic steatosis and insulin resistance in mice receiving high-fat diet.

The study is quite extensive and has been nicely performed and assembled. However, some of the molecular details need to be strengthened, particularly experiments to complete and increase the relevance of the data shown in Figures 7 and 8.

Major

1. The authors must investigate the binding of endogenous YBX1 and endogenous EDF1 with endogenous Blnc1. This should be tested using native or crosslinked RNA-protein complexes followed by IP (i.e., RIP or CLIP). It would be particularly informative if the authors could test this in liver samples (control or HFD), but it should be tested at a minimum in primary hepatocytes.
2. The interaction of EDF1 and Blnc1 should also be further examined *in vitro* by either crosslinking labeled Blnc1 followed by IP to identify bound proteins, or by analysis of recombinant purified EDF1 and YBX1 proteins and Blnc1 RNA.
3. Does Blnc1 depletion/overexpression alter the binding of endogenous LXR to endogenous target promoters (*Fasn*, *Srebp1c*), by ChIP analysis?
4. Additional molecular detail of the EDF1-Blnc1 complex are needed. The authors should map where on the body of the transcript EDF1 binds. Incidentally, how long is Blnc1 RNA?

Minor

1. In many figures, it is not immediately clear if the authors are measuring RNA or protein. From the context, the reader can eventually determine what is being assessed, but the authors are encouraged to write protein in uppercase (e.g., *FASN*, *SCD1*), and the RNA/gene in italics { } (e.g., {*Fasn*} mRNA, {*Scd1*} mRNA).

Reviewer #3 (expert in NAFLD and obesity) (Remarks to the Author):

This paper entitled "long noncoding RNA licensing of obesity-linked hepatic lipogenesis and NAFLD pathogenesis by Zhao XY et al. shows Blnc1, a long noncoding RNA previously described by the same group (in adipose tissue) contributes to the increased hepatic lipogenesis in obesity and in the development of NASH. While this paper contains potentially intriguing findings, several questions could be raised. The reviewer's comments are as follows:

1. In page 4, what is the mechanism of induction of Blnc1 expression in the liver of HFD-fed mice?
2. The authors generated global Blnc1 knockout mice. However, basal metabolic feature, and global feature of Blnc1 KO mice such as distribution of fat were not shown. Since this paper is the first description of Blnc1 KO mice, some detailed data on the mice appears to be necessary.
3. In Fig. 3b, body weight of HFD-fed liver-specific Blnc1 KO mice was reduced. Is the reduced body weight due to reduced liver weight or other organs such as adipose tissue? Changes of individual metabolic tissues are worth studying.
4. In Fig. 3d, glucose tolerance and insulin sensitivity were improved in HFD-fed liver-specific Blnc1 KO mice. Is such improvement due to reduced body weight or other mechanisms?
5. In Fig.7f, I cannot find detailed method of IP/qPCR in the text. Y-axis label of the right panel of

Fig. 7f is missing.

6. In Fig. 7e & g, it is not clearly shown which antibody was used for immunoblot analysis. Probably anti-HA was used but was not clearly shown in the figure.
7. In the absence of Blnc1, is EDF1-LXRa association decreased?
8. In the absence of LXRa (LXRa knockout), does metabolic effect of Blnc1 disappear?
9. In Fig. S1, Blnc1 expression in human tissue can be seen. Did authors have a chance to compare Blnc1 expression between lean and obese subjects?
10. In Figure 4a, the authors measured metabolic rates (VCO₂ and energy expenditure) using indirect calorimetry system. The authors showed that liver-specific Blnc1 KO (Blnc1 LKO) mice had increased VCO₂ and energy expenditure (EE). However, these analyses were performed in mice after 15 weeks of HFD feeding. As shown in Figure 3b, Blnc1 LKO had less gain of body weight by 10g compared to control mice after 10 weeks of HFD feeding. When comparing EE in animals with different body weights, analysis of covariance (ANCOVA) is recommended.
11. Liver-specific Blnc1 KO (Blnc1 LKO) mice are resistance to HFD-induced obesity, and show increased expression of thermogenic genes in BAT. However, the authors did not find the molecular mechanism underlying these phenotypes in current manuscript. As speculated by the authors, these phenotypes could be due to alteration of liver-derived endocrine hormones (such as FGF21, GDF15). Thus, it is recommended to check the level of these hormones in serum (or plasma) or liver tissues of WT and LKO mice.
12. Some more detailed description of experiment will be helpful such as AAV or Adenovirus titers, sequence of sgRNA against Blnc1, measurement of metabolic rate using indirect calorimetry or sample number for analyses of real-time PCR in several figure legends (Fig 1f, 3i or 4d), etc.

Reviewers' comments:

Reviewer #1 (expert in lipid metabolism and nuclear receptors)

The manuscript entitled "Long noncoding RNA licensing of obesity-linked hepatic lipogenesis and NAFLD pathogenesis" by Zhao et al is announced as an original work contribution. In this work the authors expand the role of the metabolic regulator lncRNA blnc1 showing that it contributes to hepatic lipogenesis and NAFLD by acting in concert with LXR and other transcriptional coactivators to induce Srebp1c expression.

The manuscript is well written and the figures are nicely rendered. Although the range of blnc1 genome-wide interactions are not well defined, on balance the work is novel, timely, and will provide an important impetus to examining mechanisms by which ligand-dependent transcription factors influence gene expression.

We thank this reviewer for his/her enthusiasm and insightful comments. We have performed several new studies to address this reviewer's comments as detailed below.

A few issues that need to be addressed

1. The LXR/Srebp1c axis is critical for hepatic responses in fasting/refeeding. Is there evidence that blnc1 is regulated during fasting and refeeding? Does loss of hepatic blnc1 impart a deficiency in hepatic responses to refeeding? Is the administration of insulin to blnc1 knockout hepatocytes sufficient to induce Srebp1c? These studies will strongly reinforce the main premise of the paper in a highly relevant physiologic context.

Hepatic lipogenesis is strongly induced by refeeding, as pointed out by the reviewer. In addition, de novo lipogenesis has also been shown to increase in response to dietary fat intake¹ and in insulin resistant state^{2,3}. We have examined Blnc1 expression during fasting and refeeding and did not observe significant changes in its mRNA levels. However, we found that Blnc1 is required for eliciting the physiological response to refeeding with regard to the induction of lipogenic genes and plasma triglyceride content (**new Supplementary Fig. 5**). We interpret these results to suggest that, while Blnc1 expression itself remains unaltered, other components of the Blnc1 transcriptional complexes may instead be nutritionally regulated.

Hepatic Blnc1 expression is elevated in insulin resistant state (Fig. 1). In our preliminary studies, we found that Blnc1 expression is induced by fatty acids and TNF α . We plan to perform detailed studies to delineate the transcription factors that mediate the induction of Blnc1 and describe our findings in a separate manuscript in future.

2. LXR activation has been shown to influence a large number of genes involved in hepatic metabolism including Abcg5, Lpcat3, rnf145, Abcg8 etc...Do the authors observe changes in canonical transcriptional targets of LXR as a consequence of loss of function and overexpression of blnc1? If the effect is restricted only to Srebp1c the authors should address this in the discussion. For example, could it be that blnc1 is deployed to only a subset of genes, are there unique coactivators that may be present at different sites, or perhaps LXR isoform specific interactions?

This reviewer brought up an excellent point. We have examined whether Blnc1 regulates a broader set of LXR target genes and found that, in addition to Srebp1c, mRNA expression of Abca1, Abcg5, Lpcat3 and Rnf145 was also induced by AAV-mediated Blnc1 overexpression (**new Fig. 1g**). In loss-of-function studies, we found that Blnc1 is

required for the induction of Srebp1c, Abca1 and Abcg5 in response to LXR agonist treatment (**Fig. 2c and Supplementary Fig. 4b**). These results indicate that Blnc1 contributes to the transcriptional function of LXR beyond its effects on Srebp1c expression.

3. Very little information is provided on the characterization of the blnc1 transcript itself in liver. What is the copy number of blnc1 in hepatocytes or liver? In addition the authors should provide single molecule RNA FISH evidence that overexpressed biotin tagged blnc1 is localized to nucleus and ideally show that it colocalizes at Srebp1c genomic locus.

As we described in previous studies, the Blnc1 transcript is approximately 965 nucleotides in length and is conserved between mice and humans^{4,5}. Blnc1 is localized to both cytosolic and nuclear compartments with higher levels in the nucleus. We have examined subcellular localization of Blnc1 in the liver and obtained similar results (**new Supplementary Fig. 2b**). We performed Blnc1 copy number measurements using qPCR with *in vitro* transcribed Blnc1 RNA as standard and estimated that approximately 120 copies of Blnc1 RNA transcripts are present in each hepatocyte.

We are in the process of performing RNA chromatin immunoprecipitation and sequencing studies to assess global chromatin occupancy by Blnc1. We also have obtained hnRNPU flox/flox mice (from Dr. Tom Maniatis lab) and are in the process of generating liver-specific hnRNPU knockout mice to examine how hnRNPU modulates Blnc1 interaction with its chromatin targets. We plan to report our findings on this topic as soon as our studies are completed.

4. Protein blots of nuclear and membrane form for Srebp1c and Srebp2 isoform should be provided.

We have performed immunoblotting of both precursor and nuclear isoform of SREBP1 on liver lysates, as shown in **new Fig. 1f, 2d, 4c, and 7d**. Srebp2 mRNA expression was unaltered by Blnc1 inactivation.

5. Please provide cholesterol levels in serum and liver for data in figures 1 and 2. Per reviewer's request, we have incorporated plasma and liver cholesterol data in **Fig. 1e and Supplementary Fig. 4a**.

6. One of the strengths of this work is use of novel approaches to generating various knockout models which would be of interest to the wider scientific community however detailed methods are not provided. For example specify name of vector used and precise cloning strategy as well as genetic background of all mice. Please provide these additional details in methods.

We have provided additional details in methods regarding the Blnc1 sgRNA transgenic vector including the sequence for sgRNAs used.

7. Similarly, the source data is not described for human tissue panel in supplementary figure 1a as well as data in figure 1b.

Blnc1 expression shown in Fig. 1b was analyzed using HFD-fed mouse liver samples. Human tissue RNA panel in Supplementary Fig. 1a was purchased from Clontech.

Reviewer #2 (expert in lncRNAs)

In this interesting manuscript, Zhao and colleagues report that the long noncoding RNA Blnc1 is part of the LXR transcriptional complex required for SREBP1c induction and hepatic lipogenic activation in obesity. Importantly, liver-specific ablation of Blnc1 prevented hepatic steatosis and insulin resistance in mice receiving high-fat diet. The study is quite extensive and has been nicely performed and assembled. However, some of the molecular details need to be strengthened, particularly experiments to complete and increase the relevance of the data shown in Figures 7 and 8.

We thank this reviewer for his/her enthusiasm and insightful comments. We have performed several new studies to address the points raised, as described below.

Major

1. The authors must investigate the binding of endogenous YBX1 and endogenous EDF1 with endogenous Blnc1. This should be tested using native or crosslinked RNA-protein complexes followed by IP (i.e., RIP or CLIP). It would be particularly informative if the authors could test this in liver samples (control or HFD), but it should be tested at a minimum in primary hepatocytes.

To confirm that endogenous Blnc1 forms ribonucleoprotein complexes with YBX1 and EDF2, we performed IP/qPCR studies using total liver lysates. As shown in **new Fig. 8c**, endogenous Blnc1 is readily detectable in immunocomplexes precipitated using anti-YBX1 or anti-EDF1 antibody. These results provide further evidence that supports the formation of ribonucleoprotein complexes between endogenous Blnc1 and EDF1 and YBX1 in the liver.

2. The interaction of EDF1 and Blnc1 should also be further examined *in vitro* by either crosslinking labeled Blnc1 followed by IP to identify bound proteins, or by analysis of recombinant purified EDF1 and YBX1 proteins and Blnc1 RNA.

We agree with this reviewer that it is important to gain additional molecular insights into how Blnc1/EDF1 ribonucleoprotein complexes to regulate gene transcription. Toward this, we have initiated studies to purify the Blnc1 and EDF1 ribonucleoprotein complexes from mouse liver using biotin aptamer-tagged Blnc1 and Flag-HA tagged EDF1. We also plan to perform Cryo-EM studies for Blnc1/EDF1 and Blnc1/YBX1 interaction in collaboration with structural biologists at the University of Michigan.

3. Does Blnc1 depletion/overexpression alter the binding of endogenous LXR to endogenous target promoters (Fasn, Srebp1c), by ChIP analysis?

We have explored whether Blnc1 inactivation affects the recruitment of LXR to its targets using ChIP assay. Compared to control, LXR occupancy on the Srebp1c promoter is markedly reduced in hepatocytes lacking Blnc1. These observations suggest that the association between Blnc1 and LXR may modulate the recruitment of LXR to its chromatin targets and its transcriptional function. The results have been incorporated in **new Fig. 9h**.

4. Additional molecular detail of the EDF1-Blnc1 complex are needed. The authors should map where on the body of the transcript EDF1 binds. Incidentally, how long is Blnc1 RNA?

As we previously reported, mouse Blnc1 is approximately 965 nucleotides in length⁵. We have generated various Blnc1 truncation mutants and tested their interaction with LXR α and EDF1. We observed that RNA domains 2 and 3 (RD2 and RD3) are essential for its interaction with LXR α and EDF1 (**new Fig. 9a-b**). In contrast, RD1 appears to be dispensable in this assay. Together with our previous studies showing that RD1 is required for the interaction between Blnc1 and hnRNPU⁴, our data support a modular domain structure for Blnc1 where different RNA domains are responsible for association with different protein factors.

Minor

1. In many figures, it is not immediately clear if the authors are measuring RNA or protein. From the context, the reader can eventually determine what is being assessed, but the authors are encouraged to write protein in uppercase (e.g., FASN, SCD1), and the RNA/gene in italics { } (e.g., {Fasn} mRNA, {Scd1} mRNA).

We have made changes throughout the text and figures to clarify this issue. All proteins are now indicated by symbols in UPPERCASE, while mRNA transcripts are indicated by *italicized* gene symbols.

Reviewer #3 (expert in NAFLD and obesity)

This paper entitled “long noncoding RNA licensing of obesity-linked hepatic lipogenesis and NAFLD pathogenesis by Zhao XY et al. shows Blnc1, a long noncoding RNA previously described by the same group (in adipose tissue) contributes to the increased hepatic lipogenesis in obesity and in the development of NASH. While this paper contains potentially intriguing findings, several questions could be raised. The reviewer’s comments are as follows:

1. In page 4, what is the mechanism of induction of Blnc1 expression in the liver of HFD-fed mice?

We have examined the effects of various metabolites and hormones on Blnc1 expression in hepatocytes and found that Blnc1 expression was induced by fatty acids and TNF α . As such, it is possible that increased hepatic lipid flux as a result of adipose tissue dysfunction may contribute to elevated Blnc1 levels in insulin resistant states. We plan to perform detailed studies to delineate the transcription factors that mediate the induction of Blnc1 and describe our findings in a separate manuscript in future.

2. The authors generated global Blnc1 knockout mice. However, basal metabolic feature, and global feature of Blnc1 KO mice such as distribution of fat were not shown. Since this paper is the first description of Blnc1 KO mice, some detailed data on the mice appears to be necessary.

Because Blnc1 is also highly expressed in adipose tissue, the metabolic phenotype in whole body knockout mice is difficult to interpret. As such, we have not performed extensive metabolic phenotype analyses using the total knockout model. We are in the process of generating and analyzing fat-specific Blnc1 knockout mice and will report the role of adipocyte Blnc1 in adipose tissue metabolism and whole body physiology in near future.

3. In Fig. 3b, body weight of HFD-fed liver-specific Blnc1 KO mice was reduced. Is the

reduced body weight due to reduced liver weight or other organs such as adipose tissue? Changes of individual metabolic tissues are worth studying.

We found that brown fat weight is lower in HFD-fed Blnc1 LKO mice compared to control, likely as a result of decreased fat content. In contrast, epididymal white fat weight appeared largely unaffected by hepatic Blnc1 inactivation. We have incorporated these data in **Supplementary Fig 7b**.

4. In Fig. 3d, glucose tolerance and insulin sensitivity were improved in HFD-fed liver-specific Blnc1 KO mice. Is such improvement due to reduced body weight or other mechanisms?

We showed that Blnc1 LKO mice had enhanced brown fat thermogenesis and reduced hepatic fat content, both of which may contribute to improved insulin sensitivity.

5. In Fig.7f, I cannot find detailed method of IP/qPCR in the text. Y-axis label of the right panel of Fig.7f is missing.

We fixed the Y-axis label in **new Fig. 9d**.

6. In Fig. 7e & g, it is not clearly shown which antibody was used for immunoblot analysis. Probably anti-HA was used but was not clearly shown in the figure.

Yes, we used anti-HA antibody for the immunoblots. We added this information in the **new Fig. 8e and 9c**.

7. In the absence of Blnc1, is EDF1-LXRa association decreased?

We performed co-IP studies in control and Blnc1 KO hepatocytes. Our new data indicate that physical interaction between LXR and EDF1 is strongly reduced in the absence of Blnc1. These results have been incorporated in **new Fig. 9f**.

8. In the absence of LXRa (LXRa knockout), does metabolic effect of Blnc1 disappear?

To address this, we performed studies in cultured hepatocytes without or with SR9238, a highly potent and selective LXR inverse agonist. We found that the ability of Blnc1 to induce Srebp1c and Fasn expression is diminished by SR9238 (**new Fig. 2g-h**), suggesting that LXR activity is required for the transcriptional effects of Blnc1 on the lipogenic pathway.

9. In Fig. S1, Blnc1 expression in human tissue can be seen. Did authors have a chance to compare Blnc1 expression between lean and obese subjects?

This is an important question. Unfortunately, we have been unable to find suitable patient samples for this purpose yet. We have initiated a collaboration with Dr. Xiaoying Li, a clinical investigator in China, to explore this question. As the liver biopsies are extremely difficult to obtain, we plan to first compare Blnc1 expression between healthy control and NASH patients.

10. In Figure 4a, the authors measured metabolic rates (VCO₂ and energy expenditure) using indirect calorimetry system. The authors showed that liver-specific Blnc1 KO

(Blnc1 LKO) mice had increased VCO₂ and energy expenditure (EE). However, these analyses were performed in mice after 15 weeks of HFD feeding. As shown in Figure 3b, Blnc1 LKO had less gain of body weight by 10g compared to control mice after 10 weeks of HFD feeding. When comparing EE in animals with different body weights, analysis of covariance (ANCOVA) is recommended.

We appreciate the reviewer bringing up this point. Indeed, there have been new considerations on how to most appropriately present the metabolic rate and energy expenditure data, as discussed in Tschop et al. and Kaiyala et al. articles^{6,7}. To date, no perfect way has been universally accepted to best report metabolic rate changes in mice. In the Blnc1 LKO mice, we found that they are partially protected from diet-induced obesity despite having elevated food intake. Based on the laws of thermodynamics, these findings suggest that the LKO mice likely have augmented energy expenditure to produce this phenotype. Per reviewer' comment, we performed ANCOVA and considered body weight as a covariate. This analysis, however, indicates that there is no statistically significant difference in VO₂ and EE between control and LKO groups. This conclusion contradicts the basic principles of thermodynamics. As such, we believe that the current presentation, while not perfect, is more consistent with the phenotypic observations in these mice. The reviewer's point has been well taken.

11. Liver-specific Blnc1 KO (Blnc1 LKO) mice are resistance to HFD-induced obesity, and show increased expression of thermogenic genes in BAT. However, the authors did not find the molecular mechanism underlying these phenotypes in current manuscript. As speculated by the authors, these phenotypes could be due to alteration of liver-derived endocrine hormones (such as FGF21, GDF15). Thus, it is recommended to check the level of these hormones in serum (or plasma) or liver tissues of WT and LKO mice.

We have measured hepatic expression of Fgf21 and Gdf15 and found that their mRNA levels were comparable between control and Blnc1 LKO mouse livers. We incorporated this information in **Supplementary Fig. 7f** and the text.

12. Some more detailed description of experiment will be helpful such as AAV or Adenovirus titers, sequence of sgRNA against Blnc1, measurement of metabolic rate using indirect calorimetry or sample number for analyses of real-time PCR in several figure legends (Fig 1f, 3i or 4d), etc.

We thank this reviewer for the suggestions and have included additional details of these studies in the Methods section and relevant figure legends.

References:

1. Lin J, *et al.* Hyperlipidemic effects of dietary saturated fats mediated through PGC-1beta coactivation of SREBP. *Cell* **120**, 261-273 (2005).
2. Kohjima M, *et al.* SREBP-1c, regulated by the insulin and AMPK signaling pathways, plays a role in nonalcoholic fatty liver disease. *Int J Mol Med* **21**, 507-511 (2008).

3. Shimomura I, Bashmakov Y, Horton JD. Increased levels of nuclear SREBP-1c associated with fatty livers in two mouse models of diabetes mellitus. *The Journal of biological chemistry* **274**, 30028-30032 (1999).
4. Mi L, Zhao XY, Li S, Yang G, Lin JD. Conserved function of the long noncoding RNA Blnc1 in brown adipocyte differentiation. *Mol Metab* **6**, 101-110 (2017).
5. Zhao XY, Li S, Wang GX, Yu Q, Lin JD. A long noncoding RNA transcriptional regulatory circuit drives thermogenic adipocyte differentiation. *Mol Cell* **55**, 372-382 (2014).
6. Kaiyala KJ, Schwartz MW. Toward a more complete (and less controversial) understanding of energy expenditure and its role in obesity pathogenesis. *Diabetes* **60**, 17-23 (2011).
7. Tschop MH, *et al.* A guide to analysis of mouse energy metabolism. *Nat Methods* **9**, 57-63 (2011).

REVIEWERS' COMMENTS:

Reviewer #1 (Remarks to the Author):

The authors have added substantial new data supporting the main conclusions of the paper and I am supportive of publication.

A minor note for the final version of the manuscript

The authors should include detailed data on the copy number calculation and figure including primer sequences used (As an example of RNA copy number figure please see PMID: 27525555). The 120 copy numbers per cell seems remarkably high and would make Blnc1 one of the most abundant lncRNAs? Are the authors sure they are not detecting transcription from neighboring gene as well? Public genome browser annotations suggest overlap between blnc1 and isoforms of its neighboring protein coding gene.

Reviewer #2 (Remarks to the Author):

The authors have provided thoughtful responses to my concerns.

Reviewer #3 (Remarks to the Author):

This revised manuscript by Zhao XY et al. has been improved by incorporation of the reviewer's comments.

The answers to the question 1 and 2 of the reviewer 3 are understandable. However, incorporation of at least parts of the authors' response to the question 1 and 2 into the text is recommended (e.g. into the Discussion section).

REVIEWERS' COMMENTS:

Reviewer #1 (Remarks to the Author):

The authors have added substantial new data supporting the main conclusions of the paper and I am supportive of publication.

A minor note for the final version of the manuscript.

The authors should include detailed data on the copy number calculation and figure including primer sequences used (As an example of RNA copy number figure please see PMID: 27525555). The 120 copy numbers per cell seems remarkably high and would make Blnc1 one of the most abundant lncRNAs? Are the authors sure they are not detecting transcription from neighboring gene as well? Public genome browser annotations suggest overlap between blnc1 and isoforms of its neighboring protein coding gene.

Blnc1 is indeed a relatively abundant lncRNA in the liver. Its qPCR Ct value is approximately 20, which makes it a highly expressed gene. We have included the copy number data in the text and Supplementary Figure 1b. With regard to Blnc1 gene structure, we have performed RACE experiments and conclusively established the 5' and 3' ends of Blnc1 transcript, as described in the Zhao et al. paper in Molecular Cell in 2014.

Reviewer #2 (Remarks to the Author):

The authors have provided thoughtful responses to my concerns.

Reviewer #3 (Remarks to the Author):

This revised manuscript by Zhao XY et al. has been improved by incorporation of the reviewer's comments.

The answers to the question 1 and 2 of the reviewer 3 are understandable. However, incorporation of at least parts of the authors' response to the question 1 and 2 into the text is recommended (e.g. into the Discussion section).

We have incorporated this information in the Results and Discussion sections on pages 6 and 14, respectively.